# Markov Equivalence and Consistency in Differentiable Structure Learning

**Chang Deng**[†][*]  **Kevin Bello**[†,‡]  **Pradeep Ravikumar**[‡]  **Bryon Aragam**[†]

[†]Booth School of Business, University of Chicago, Chicago, IL 60637
[‡]Machine Learning Department, Carnegie Mellon University, Pittsburgh, PA 15213

## Abstract

Existing approaches to differentiable structure learning of directed acyclic graphs (DAGs) rely on strong identifiability assumptions in order to guarantee that global minimizers of the acyclicity-constrained optimization problem identifies the true DAG. Moreover, it has been observed empirically that the optimizer may exploit undesirable artifacts in the loss function. We explain and remedy these issues by studying the behavior of differentiable acyclicity-constrained programs under general likelihoods with multiple global minimizers. By carefully regularizing the likelihood, it is possible to identify the sparsest model in the Markov equivalence class, even in the absence of an identifiable parametrization. We first study the Gaussian case in detail, showing how proper regularization of the likelihood defines a score that identifies the sparsest model. Assuming faithfulness, it also recovers the Markov equivalence class. These results are then generalized to general models and likelihoods, where the same claims hold. These theoretical results are validated empirically, showing how this can be done using standard gradient-based optimizers (without resorting to approximations such as Gumbel-Softmax), thus paving the way for differentiable structure learning under general models and losses. Open-source code is available at https://github.com/duntrain/dagrad.

## 1 Introduction

Directed acyclic graphs (DAGs) are the most common graphical representation for causal models [48, 60, 50], where nodes represent variables and directed edges represent cause-effect relationships among variables. We are interested in the problem of structure learning, i.e. learning DAGs from passively observed data, also known as causal discovery. Our focus will mainly be on score-based approaches to DAG learning [11, 23], where the structure learning problem is formulated as optimizing a given score or loss function $s(B; \mathbf{X})$ that measures how well the graph, represented as an adjacency matrix $B \in \{0,1\}^{p \times p}$, fits the observed data $\mathbf{X}$, constrained to the graphical structure $B$ being acyclic. This combinatorial optimization problem is known to be NP-complete [10, 12].

Recent advances in score-based methods have introduced a continuous representation of DAGs, transforming the combinatorial acyclicity constraint into a continuous constraint via a differentiable function that exactly characterizes DAGs [73]. In this case, the discrete adjacency matrix $B \in \{0,1\}^{p \times p}$ is first relaxed to the space of real matrices, i.e., $B \in \mathbb{R}^{p \times p}$, and then a differentiable function $h : \mathbb{R}^{p \times p} \to [0, \infty)$ is devised so that $h(B) = 0$ if and only if $B$ is a DAG [73, 4]. This results in the following optimization problem:

$$\min_{B \in \mathbb{R}^{p \times p}} s(B; \mathbf{X}) \quad \text{subject to} \quad h(B) = 0. \tag{1}$$

Considering a differentiable score function $s$, the differentiable program (1) facilitates the use of gradient-based optimization techniques along with the use of richer models, such as neural networks,

---

[*]changdeng@chicagobooth.edu

38th Conference on Neural Information Processing Systems (NeurIPS 2024).

for modeling the functional relationships among the variables [74, 67, 41, 29, 44, 28, 76]. One of the most attractive features of this approach is that it applies to general models, losses, and optimizers, in contrast to prior work. Moreover, it cleanly separates computational and statistical concerns, so that each can be studied in isolation, in the same spirit as the graphical lasso [36, 68, 17].

Looking back at the inception of the continuous DAG learning framework by Zheng et al. [73, 74], however, most developments in this framework have focused on the design of alternative differentiable acyclicity functions $h$ with better numerical/computational properties [4, 32, 72, 67], placing little emphasis on which score function to use [41]. In fact, and unfortunately, regardless of the modeling assumptions, it has become a rather standard practice [67, 74, 4, 13, 32, 28] to simply use the least squares (LS) loss (a.k.a. "reconstruction loss") as the score by default, following the original paper by Zheng et al. [73], despite its known statistical limitations [63, 33, 1].

As a result, Reisach et al. [55] flagged the empirical successes of continuous structure learning (CSL) methods as largely due to the high agreement between the order of marginal variances of the nodes and the topological order of the underlying simulated DAGs, a concept they describe as "varsortability". Then, Reisach et al. [55] empirically showed that the performance in structure recovery of CSL methods drops significantly after simple data standardization. More recently, Ng et al. [42] demonstrated that this phenomenon may not be explained by varsortability, and instead pointed out that the explanations are due to the score function, albeit without proposing which score function to use. These observations motivate a deeper consideration of the choice of score.

Unfortunately, despite the fact that several score functions have been proposed for learning Bayesian networks (such as BIC [23], BDeu [35], and MDL [6]), their application to CSL methods is not well understood. This paper is precisely concerned with finding a suitable and general score function with strong statistical properties for CSL methods. That is, our objective is to find a score function that is: (1) intrinsically differentiable so that it is amenable to gradient-based optimization without approximations; (2) applicable to general models; (3) scale-invariant; (4) capable of identifying the sparsest model under proper regularization; and (5) connects nicely with classical concepts from Bayesian networks such as faithfulness and Markov equivalence classes.

**Contributions.** The main contribution of our work is to show that a properly regularized, likelihood-based score function has the five properties outlined above. We begin with Gaussian models to convey the main ideas, and then discuss generalizations. In more detail:

1. (Section 4) Starting with Gaussian models, we show that using the log-likelihood with a quasi-MCP penalty (10) as the scoring function leads to optimal solutions of (1) that correspond to the sparsest DAG structure which is Markov to $P(X)$ (Theorem 1). Furthermore, under the faithfulness assumption, all optimal solutions are the sparsest within the same Markov equivalence class (Theorem 2).

2. (Section 5) We provide general conditions on the log-likelihood under which similar results hold for general models (Theorem 4).

3. (Section 4.5) We show that for Gaussian models, the log-likelihood score is scale-invariant. This means that rescaling or standardizing the data does not change the DAG structure (Theorem 3), and hence is not susceptible to varsortability.

4. We conduct experiments to evaluate the advantages of using a likelihood-based score. The findings from these experiments are detailed in in Section 6. The empirical results support our theory: The likelihood-based score is robust and scale invariant. We also release open-source code to facilitate the implementation and reproduction of our results.

## 2 Related work

Most methods for learning DAGs fall into two primary categories: Constraint-based algorithms, which depend on tests of conditional independence, and score-based algorithms, which aim to optimize a specific score or loss function. As our focus is on score-based methods, we only briefly mention classical constraint-based methods [59, 34, 62]. Within the umbrella of score-based methods, the linear Gaussian models is covered in works such as [1, 2, 19, 20, 37, 49], while studies on linear non-Gaussian SEMs are found in [33, 57]. Regarding nonlinear SEMs, significant contributions have been made in additive models [9, 15, 64], additive noise models [24, 49, 39], generalized linear models [47, 46, 22], and broader nonlinear SEMs [38, 21].

Works that are more directly connected to our research include those developed in the continuous structure learning (CSL) framework [e.g. 73, 74, 13, 4, 14, 29, 76, 41, 40, 28, 44]. Most of these papers focus on empirical and computational aspects, and only a few study the theoretical properties of the CSL framework in (1). These include: [65, 43] studied the optimization and convergence subtleties of problem (1); [13] studied optimality guarantees for more general types of score functions and proposed a bi-level optimization method to guarantee local minima; [14] designed an optimization scheme that converges to the global minimum of the least squares score in the bivariate case. Finally, among the few works that study score functions under this framework, we note: [41] studied the properties of the $\ell_1$-regularized profile log-likelihood, which leads to quasi-equivalent models to the ground-truth DAG; and the authors in [56] claim that a family of likelihood-based scores reduce to the least square loss, although this only holds under knowledge of the noise variances [33]. Perhaps most closely related to our work is [8], who proved a similar identifiability result under the likelihood score. However, they used an $\ell_0$ regularizer along with the faithfulness assumption, which leads to an inherently non-differentiable optimization problem that is much simpler to analyze but requires approximations (e.g. Gumbel-Softmax) to optimize. On the other hand, they also consider interventional data, which we do not pursue in this work. Extending our results to include interventional data and interventional Markov equivalence is an important direction for future work. In contrast to the aforementioned works, we also prove that the log-likelihood has desirable properties such as being scale invariant, and when regularized by nonconvex and differentiable approximations of the $\ell_0$ function, it provably leads to useful solutions that are minimal models and Markov equivalent to the underlying structure, without assuming faithfulness.

## 3  Preliminaries

We let $G = (V, E)$ denote a directed graph on $p$ nodes, with vertex set $V = [p] := \{1, \ldots, p\}$ and edge set $E \subset V \times V$, where $(i, j) \in E$ indicates the presence of a directed edge from node $i$ to node $j$. We associate each node $i \in V$ to a random variable $X_i$, and let $X = (X_1, \ldots, X_p)$.

**Structural equation models (SEMs).** An SEM $(X, f, P(N))$ over the random vector $X = (X_1, \ldots, X_p)$ is a collection of $p$ structural equations of the form:

$$X_j = f_j(X, N_j), \quad \partial_k f_j = 0 \text{ if } k \notin \text{PA}_j, \tag{2}$$

where $f = (f_j)_{j=1}^p$ is a collection of functions $f_j : \mathbb{R}^{p+1} \to \mathbb{R}$, here $N = (N_1, \ldots, N_p)$ is a vector of independent noises with distribution $P(N)$, and $\text{PA}_j$ denotes the set of parents of node $j$. Here, $\partial_k f_j$ denotes the partial derivative of $f_j$ w.r.t. $X_k$, which is identically zero when $f_j$ is independent of $X_k$, i.e. $f_j(X, N_j) = f_j(\text{PA}_j, N_j)$. The graphical structure induced by the SEM, assumed to be a DAG, will be represented by the following $p \times p$ weighted adjacency matrix $B$:

$$B = B(f), \qquad B_{ij} = \|\partial_i f_j\|_2, \tag{3}$$

and we use $G(B)$ to denote the corresponding binary adjacency matrix. For any set $\mathcal{B}$ of SEMs, let

$$\mathcal{G}(\mathcal{B}) := \{G(B(f)) : (X, f, P(N)) \in \mathcal{B}\}, \tag{4}$$

i.e. $\mathcal{G}(\mathcal{B})$ is the collection of all the DAGs implied by $\mathcal{B}$. If $\mathcal{D}$ is a set of DAGs and $\mathcal{B}$ is a set of SEM, we also abuse notation by writing $\mathcal{D} = \mathcal{B}$ to indicate $\mathcal{D} = \mathcal{G}(\mathcal{B})$.

The SEM (2) is general enough to include many well-known models, such as linear SEMs [e.g., 33, 49], generalized linear models [47, 45, 18], and additive noise models [24, 51], post-nonlinear models [70, 71] and general nonlinear SEM [38, 21, 26, 74]. To illustrate some of these models: In linear SEMs we have $X_j = f_j(\text{PA}_j) + N_j$, where $f_j$ is a linear map; in causal additive models (CAM) we have $X_j = \sum_{k \in \text{PA}_j} f_{j,k}(X_k) + N_j$, where $f_{j,k}$ is a univariate function; in post-nonlinear models we have $X_j = f_{j,1}(f_{j,2}(\text{PA}_j) + N_j)$. In fact, essentially any distribution can be represented as an SCM of the form (2); see Proposition 7.1 in Peters et al. [50].

**Faithfulness and sparsest representations.** It is well-known that the DAG $G$ is *not* always identifiable from $X$, and there is a well-developed theory on what can be identified based on $X$ under certain assumptions. This leads to the concepts of faithfulness and sparsest representations, which we briefly recall here; we refer the reader to [60, 48, 50] for details. Let $\mathcal{I}(P)$ denote the set of conditional independence relations implied by the distribution $P$, and let $\mathcal{I}(G)$ denote the set of $d$-separations implied by the graph $G$. Then $P$ is *Markov* to $G$ if $\mathcal{I}(G) \subset \mathcal{I}(P)$, and *faithful* to $G$ if $\mathcal{I}(P) \subset \mathcal{I}(G)$. When both conditions hold, i.e. $\mathcal{I}(P) = \mathcal{I}(G)$, then $G$ is called a *perfect map* of $P$. Following common convention, we will simply call $P$ faithful when $\mathcal{I}(G) = \mathcal{I}(P)$. When $P$ is faithful to $G$, the Markov equivalence class (MEC) of $G$ is identifiable and can be represented by a CPDAG.

**Definition 1.** *For any DAG $G$, the Markov equivalence class is $\mathcal{M}(G) = \{\widetilde{G} : \mathcal{I}(\widetilde{G}) = \mathcal{I}(G)\}$*

Since faithfulness may not always hold, there has been progress in understanding what can be identified under weaker conditions. One approach which we will use is the notion of a *sparsest (Markov) representation* (SMR), introduced in [54]. A sparsest representation of $P$ is a Markovian DAG $G$ that has strictly fewer edges than any other Markovian DAG $G'$, and such sparsest representation is unique up to Markov equivalence class. Theorem 2.4 in [54] shows that if $P$ is faithful to $G$, then $G$ must be a sparsest representation of $P$. This notion is closely related to the notion of minimality we adopt in Definition 2 (cf. Lemma 4 in the Appendix). These ideas can be generalized and weakened even further; see [31, 30] for details.

**Parameters and the negative log-likelihood (NLL).** For positive integers $m, s$, we will use $\psi \in \Psi \subseteq \mathbb{R}^m$ and $\xi \in \Xi \subseteq \mathbb{R}^s$ to denote the model parameters for $f = (f_1, \ldots, f_p)$ and $N$, respectively.[2] Then we denote the distribution of $X$ by $P(X; \psi, \xi)$. Let $\mathbf{x} \in \mathbb{R}^p$ denote one observation of $X$. Given $n$ i.i.d. samples $\mathbf{X} = (\mathbf{x}_1, \ldots, \mathbf{x}_n)^\top$ where $\mathbf{x}_i \sim P(X; \psi, \xi)$, the negative log-likelihood and expected negative log-likelihood can be written as:

$$\ell_n(\psi, \xi) = -\frac{1}{n} \sum_{i=1}^n \log P(\mathbf{x}_i; \psi, \xi), \qquad \ell(\psi, \xi) = -\mathbb{E}[\log P(\mathbf{x}; \psi, \xi)], \tag{5}$$

where the subscript $n$ in $\ell_n$ is used to indicate the sample version of the log-likelihood.

**Identifiability.** Let $\psi^0$ (resp. $\xi^0$) denote the model parameters for the ground truth $f^0$ (resp. $N^0$), let $B^0 = B^0(\psi^0) \in \mathbb{R}^{p \times p}$ denote the induced weighted adjacency matrix, and let $G(B^0) \in \{0, 1\}^{p \times p}$ denote the induced binary adjacency matrix. For example, in the general linear Gaussian model (6), $\psi = B$ represents the adjacency matrix, and $\xi = \Omega$ denotes the variance of the Gaussian noise. In another case, if $f_j$ is approximated by a multilayer perceptron (MLP), with $N_j$ as Gaussian noise, then $\psi$ includes all the parameters of the MLP, while $\xi$ represents the variance of the Gaussian noise. Additionally, $(B)_{ij} = [B(\psi)]_{ij} = \|\text{i-th column of } A_j^{(1)}\|$, where $A_j^{(1)}$ is the first hidden layer in $f_j$ [74]. Thus, by our definitions, $P(X; \psi^0, \xi^0)$ is the true distribution. Here, there are two types of identifiability questions:

1. *Parameter identifiability*: Is it possible to uniquely determine the parameters $(\psi^0, \xi^0)$ based on observations from $P(X; \psi^0, \xi^0)$? Formally, is there any $(\widetilde{\psi}, \widetilde{\xi}) \neq (\psi^0, \xi^0)$, such that $P(X, \psi^0, \xi^0) = P(X, \widetilde{\psi}, \widetilde{\xi})$ almost surely?

2. *Structural identifiability*: Is it possible to uniquely determine the DAG $G(B^0)$ based on observations from $P(X; \psi^0, \xi^0)$? In other words, is there any $(\widetilde{\psi}, \widetilde{\xi}) \neq (\psi^0, \xi^0)$ such that $P(X, \psi^0, \xi^0) = P(X, \widetilde{\psi}, \widetilde{\xi})$ but $G(B^0) \neq G(B(\widetilde{\psi}))$.

In general, parameter identifiability implies structural identifiability since the ability to uniquely determine parameter values often means that the structure they induce is also identifiable. However, the converse is not generally true, i.e. structural identifiability does not always imply parameter identifiability, as different parameter values can lead to the same structure. Classical results on identifiability of SEMs include: linear SEM with equal variance [33], linear SEM with non-Gaussian noises [57, 58], causal additive models with Gaussian noises [9], additive models with continuous noise [51], and post-nonlinear models [70, 71, 25].

In models where parameter identifiability is possible, the population NLL $\ell(\psi, \xi)$ serves as a natural choice for the score function because it attains a unique minimum at the true parameters $(\psi^0, \xi^0)$. However, this approach is not straightforward for nonidentifiable models, where multiple parameter sets can induce the same data distribution $P(X; \psi^0, \xi^0)$, leading to ambiguities in parameter or structure estimation. In such cases, regularizing the log-likelihood can alleviate this issue. These regularizers enforce specific characteristics like sparsity, guiding the model towards more meaningful solutions (e.g. faithful or sparsest), despite the lack of identifiability.

## 4 General linear Gaussian SEMs: A nonidentifiable model

Although our results apply to general models, we begin by outlining the main idea with one of the simplest nonidentifiable models, the Gaussian model. Our goal in this section is to theoretically show

---

[2] Given that $\psi$ describes all the parameters for the functions $f$, we will also use $B(\psi)$ to denote $B(f)$ in (3).

how the NLL with nonconvex differentiable regularizers can lead to meaningful solutions such as minimal-edge models and elements of Markov equivalent classes. We also discuss and prove the scale invariance of NLL, making it amenable to CSL approaches and addressing concerns raised in previous work [55, 42]. Then, in Section 5, we extend these results to general models.

## 4.1 Gaussian DAG models

A linear SEM $(B, \Omega)$ over $X$ with independent Gaussian noises $N$, a special case of (2), is well-known to be nonidentifiable in terms of parameters and structure [see 2, for discussion]. We write the model as follows:

$$X = B^\top X + N, \tag{6}$$

where $B \in \mathbb{R}^{p \times p}$ is a matrix of coefficients with $G(B)$ being a DAG, and $N \in \mathbb{R}^p$ is the vector of independent noises with covariance matrix $\Omega = \mathrm{diag}(\omega_1^2, \ldots, \omega_p^2)$.[3]

Given the model (6) it is easy to see that the distribution $P(X)$ is Gaussian and is fully characterized by the pair $(B, \Omega)$. That is:

$$X \sim \mathcal{N}(0, \Sigma), \quad \Sigma = \Sigma_f(B, \Omega) \coloneqq (I - B)^{-\top} \Omega (I - B)^{-1}, \tag{7}$$

where $\Sigma$ is the covariance matrix of $X$. In the sequel, we use the subscript $f$ to refer to a function. In this case, $\Sigma_f$ denotes a function with arguments $(B, \Omega)$ and returns the covariance matrix. Moreover, we use $\Theta$ to denote the corresponding precision matrix (inverse of the covariance matrix):

$$\Theta = \Theta_f(B, \Omega) \coloneqq (I - B) \Omega^{-1} (I - B)^\top. \tag{8}$$

Let $\mathbf{X} = (\mathbf{x}_1, \ldots, \mathbf{x}_n)^\top$ be $n$ i.i.d. samples of $X$. Then, let the sample covariance matrix be $\widehat{\Sigma} = \frac{1}{n} \sum_{i=1}^n \mathbf{x}_i \mathbf{x}_i^\top$. The sample NLL function is given by:

$$\ell_n(B, \Omega) = -\frac{1}{n} \log \prod_{i=1}^n P(\mathbf{x}_i; B, \Omega) = \frac{1}{2} \log \det \Omega - \log \det(I - B) + \frac{1}{2} \mathrm{Tr}(\widehat{\Sigma} \Theta(B, \Omega)) + \mathrm{const.}$$

The corresponding population NLL function is

$$\ell(B, \Omega) = -\mathbb{E}_X \log P(X; B, \Omega) = \frac{1}{2} \log \det \Omega - \log \det(I - B) + \frac{1}{2} \mathrm{Tr}(\Sigma \Theta(B, \Omega)) + \mathrm{const.}$$

The full derivation can be found in Appendix C.1. Here, it is important to note that the distribution of $X$ is fully determined by either the precision matrix $\Theta$ or the covariance matrix $\Sigma$.

## 4.2 Equivalence and nonidentifiability

Our goal is to identify $(B, \Omega)$: Unfortunately, the model is inherently nonidentifiable in terms of both parameter and structure. This means that multiple pairs $(B, \Omega)$ for model (6) can induce the same data distribution $P(X)$ given in (7), thus resulting also in the same precision matrix $\Theta$. To address this, we define the equivalence class $\mathcal{E}(\Theta)$ as the set of all pairs $(B, \Omega)$ such that $\Theta_f(B, \Omega) = \Theta$.

$$\mathcal{E}(\Theta) \coloneqq \{(B, \Omega) : \Theta_f(B, \Omega) = \Theta\}. \tag{9}$$

It is worth noting that the size of $\mathcal{E}(\Theta)$ is finite and at most $p!$, which corresponds to the number of permutations for $p$ variables [2]. For more comprehensive details on this class, see Appendix C.2.

This ambiguity naturally leads to the question: which pair $(B, \Omega)$ should we estimate? Since any pair would be indistinguishable based only on observational data, a natural objective is to estimate the "simplest" DAG, for example, a DAG that induces the precision matrix $\Theta$ with the smallest number of edges. In other words, our goal is to estimate the matrix $B$ that has the minimal number of nonzero entries in the equivalence class. Let $s_B = |\{(i, j) : B_{ij} \neq 0\}|$.

**Definition 2** (Minimality). *$(B, \Omega)$ is called a minimal-edge I-map[4] in the equivalence class $\mathcal{E}(\Theta)$ if $s_B \leq s_{\widetilde{B}}, \forall (\widetilde{B}, \widetilde{\Omega}) \in \mathcal{E}(\Theta)$. The set of all minimal-edge I-maps in the equivalence class $\mathcal{E}(\Theta)$ is referred to as the minimal equivalence class $\mathcal{E}_{\min}(\Theta)$:*

$$\mathcal{E}_{\min}(\Theta) = \{(B, \Omega) : (B, \Omega) \text{ is minimal-edge I-map}, (B, \Omega) \in \mathcal{E}(\Theta)\}.$$

---

[3]In terms of notation given in Section 3, we have parameters $\psi = B$ and $\xi = (\omega_1^2, \ldots, \omega_p^2)$, and parameter spaces $\Psi = \mathbb{R}^{p \times p}, \Xi = \mathbb{R}_{>0}^p$.

[4]This generalizes the classical definition for DAGs [e.g. 63] to refer to the entire model with the distribution and graph encoded by the matrix $B$ and the error variance $\Omega$.

In the sequel, for brevity, we will often refer to such models as "minimal models".

Unlike faithfulness, which may not always hold, the minimal equivalence class $\mathcal{E}_{\min}(\Theta)$ is always well-defined. Moreover, as detailed in Lemma 4 in the Appendix, Definition 2 is closely related to the SMR assumption [54]: Under the SMR assumption (and hence also faithfulness) for $G$, we have $\mathcal{M}(G) = \mathcal{E}_{\min}(\Theta)$, i.e., $\mathcal{E}_{\min}(\Theta)$ is the Markov equivalence class of $G$. However, there could be multiple pairs $(B, \Omega)$ within $\mathcal{E}_{\min}(\Theta)$. Nevertheless, our goal is to recover one element from $\mathcal{E}_{\min}(\Theta)$. The elements in $\mathcal{E}_{\min}(\Theta)$ not only represent the "simplest" DAG model for $X$ in terms of edge count, but also bear a deep connection to classical notions such as Markov equivalence. For example, under faithfulness, all these elements describe the same independence statements.

**Lemma 1.** *Let $X$ follow model* (6) *with* $(B^0, \Omega^0)$ *and* $\Theta^0 = \Theta_f(B^0, \Omega^0)$. *Assume that $P(X)$ is faithful to $G^0 \coloneqq G(B^0)$. Then $\mathcal{M}(G^0) = \mathcal{E}_{\min}(\Theta^0)$.*

Recall our convention that this means that $\mathcal{M}(G^0) = \mathcal{G}(\mathcal{E}_{\min}(\Theta^0))$, i.e. the DAG structures contained in $\mathcal{E}_{\min}(\Theta^0)$ coincide with $\mathcal{M}(G^0)$. Thus, under the faithfulness assumption, recovering $\mathcal{E}_{\min}(\Theta^0)$ is the same as recovering the MEC, which is the usual goal in causal discovery. Moreover, we emphasize that these apply generally: For non-Gaussian $X$ following the model specified in (2), the same conclusion can be made; see Lemma 3 in the Appendix. Finally, we note that the commonly used LS loss does *not* have the same minimizers as the log-likelihood when the noise variances are different; see Appendix C.3.

## 4.3 Regularization

In order to distinguish elements in $\mathcal{E}(\Theta)$ from the minimal elements in $\mathcal{E}_{\min}(\Theta)$, we need to somehow account for the number of edges when evaluating the score function. The common approach to this is to use BIC, or equivalently the $\ell_0$ penalty. Although both approaches effectively penalize the number of nonzero entries in $B$, their non-differentiability makes them unsuitable for *differentiable* structure learning. The $\ell_1$ penalty, while amenable to differentiable approaches,[5] is not effective in precisely counting the number of edges, and also biased in parameter estimation[6]. To mitigate these shortcomings alternatives such as the smoothly clipped absolute deviation (SCAD) penalty [16] and the minimax concave penalty (MCP) [69] have been proposed. We choose to use a reparametrized version of MCP, termed quasi-MCP, defined as follows:

$$\text{quasi-MCP:} \qquad p_{\lambda, \delta}(t) = \lambda[(|t| - \frac{t^2}{2\delta})\mathbb{1}(|t| < \delta) + \frac{\delta}{2}\mathbb{1}(|t| > \delta)] \tag{10}$$

Here, $\mathbb{1}(\cdot)$ is the indicator function; corresponding plot can be found in Appendix C.5. Similar to MCP, quasi-MCP is a symmetric function that takes on a quadratic form between $[0, \delta]$ and remains constant for values greater than $\delta$. The function is smooth, and for values $|t| > \delta$, it approximates the behavior of the $\ell_0$ penalty, thus serving to penalize the number of non-zero coefficients in $B$.

The score function in (1) can be naturally written as

$$s(B, \Omega; \lambda, \delta, \mathbf{X}) = \ell_n(B, \Omega) + p_{\lambda, \delta}(B) \tag{11}$$

where $p_{\lambda, \delta}(B) = \sum_{i \neq j} p_{\lambda, \delta}(B_{ij})$. Then, the optimization problem can be written as

$$\min_{B, \Omega} s(B, \Omega; \lambda, \delta, \mathbf{X}) \quad \text{subject to} \quad h(B) = 0, \ \Omega > 0. \tag{12}$$

It is worth noting that for any $B$, the corresponding optimal $\Omega$ that minimizes $s(B, \Omega; \lambda, \delta, \mathbf{X})$ can be easily be expressed in terms of $B$ as $\Omega_f(B)$ (see Appendix C.1). Therefore, we can always plug $\Omega_f(B)$ into (12) to profile out $\Omega$.

## 4.4 Provably recovering minimal models

Solving problem (12) requires minimizing $\ell_n(B, \Omega)$ and $p_{\lambda, \delta}$ simultaneously. To study the behavior of these minimizers, let us define the set of global minimizers,

$$\mathcal{O}_{n, \lambda, \delta} = \{(B^*, \Omega^*) : (B^*, \Omega^*) \text{ is a minimizer of (12)}\}. \tag{13}$$

---

[5]Although $\ell_1$ is nonsmooth, standard smoothing techniques can be applied to $\ell_1$ regularizers as in [73, 4].
[6]See Appendix C.4 for examples.

Ideally, we would like $\mathcal{O}_{n,\lambda,\delta} = \mathcal{E}_{\min}(\Theta^0)$, however, it is unclear whether there exist values of $\lambda$ and $\delta$ such that any optimal solution $(B^*, \Omega^*)$ lies within $\mathcal{E}_{\min}(\Theta)$. The following theorem provides an affirmative answer to this question. In the sequel, we say that a property $S(x)$ holds for all sufficiently small $x > 0$ if there is some fixed $\epsilon > 0$ such that for every $x \le \epsilon$, the property $S(x)$ holds.

**Theorem 1.** *Let $X$ follow model* (6) *with* $(B^0, \Omega^0)$ *and* $\Theta^0 = \Theta_f(B^0, \Omega^0)$. *Let* $\mathbf{X}$ *be $n$ i.i.d. samples from $P(X)$, and $\mathcal{O}_{n,\lambda,\delta}$ be defined as in* (13). *Then, for all sufficiently small $\lambda, \delta > 0$ (independent of $n$), it holds that* $P(\mathcal{O}_{n,\lambda,\delta} = \mathcal{E}_{\min}(\Theta^0)) \to 1$ *as $n \to \infty$.*

In other words, we can always guarantee that $\mathcal{O}_{n,\lambda,\delta} = \mathcal{E}_{\min}(\Theta^0)$ by taking $\lambda, \delta$ sufficiently small, which is easily accomplished in practice. In the following, we use the superscript $0$ to denote ground truth parameters. Additionally, we can assume that $B^0$ always belongs to $\mathcal{E}_{\min}(\Theta^0)$, ensuring that our reference to the ground truth aligns with the simplest or minimal representation within the equivalence class. Moreover, by Lemma 1, under the faithfulness assumption, Theorem 1 can be interpreted as recovering the Markov equivalence class $\mathcal{M}(G^0)$:

**Theorem 2.** *Consider the setup in Theorem 1 and assume additionally that $P(X)$ is faithful to $G^0 \coloneqq G(B^0)$. Then, for all sufficiently small $\lambda, \delta > 0$ (independent of $n$), it holds that* $P(\mathcal{O}_{n,\lambda,\delta} = \mathcal{M}(G^0)) \to 1$ *as $n \to \infty$.*

Theorem 2 indicates with properly chosen hyperparameters, the optimal solution from optimization (12) will produce a graph that adheres to the same independence statements as $G^0$. This implies that the structure learned through the optimization process accurately reflect the underlying causal or conditional independence structure of underlying data generating process.

**Remark 1.** *Although we use quasi-MCP (mainly for its simplicity), it turns out MCP or SCAD can also be used. See Corollary 1 in Appendix A for details.*

## 4.5 Scale invariance and standardization

It is known that the LS loss is not *scale-invariant*, i.e. re-scaling the data (and in particular, standardizing it) can drastically change the structure [33], a fact which Reisach et al. [55] use to argue that differentiable DAG learning with the LS Loss is also not scale-invariant. Here we show that by using a different score—in this case the log-likelihood—fixes this and results in (provable) scale invariance. Thus, the choice of score function is crucial if certain properties such as scale invariance are desired. The following result restates the well-known fact that Gaussian DAGs are invariant to re-scaling (i.e. re-scaling does not change the support for any $(B, \Omega) \in \mathcal{E}(\Theta)$) using our notation:

**Lemma 2.** *Let $X \sim \mathcal{N}(0, \Sigma)$, suppose $\Sigma$ is a positive definite covariance matrix and let $\Theta \coloneqq \Sigma^{-1}$, suppose $D$ is a diagonal matrix with positive diagonal entries. Then $\mathcal{G}(\mathcal{E}(\Theta)) = \mathcal{G}(\mathcal{E}(D\Theta D))$.*

Lemma 2 has appealing consequences for standardization. Given raw data $\mathbf{X}$, denote its standardized version by $\mathbf{Z}$ (cf. Appendix C.6). Ideally, structure learning algorithms will output the same structure whether $\mathbf{X}$ or $\mathbf{Z}$ is used as input, and Lemma 2 suggests that re-scaling $\mathbf{X}$ will not alter the structure of the DAG that is recovered from optimizing (12). The following theorem formalizes this:

**Theorem 3.** *Under the same setting as Theorem 1, the solutions to* (12) *are scale-invariant. That is, for any $n \ge 0$, let*

$$\mathcal{O}_{n,\lambda,\delta}(\mathbf{X}) = \{(B^*, \Omega^*) : (B^*, \Omega^*) \text{ is a minimizer of (12) with data } \mathbf{X}\},$$
$$\mathcal{O}_{n,\lambda,\delta}(\mathbf{Z}) = \{(B^*, \Omega^*) : (B^*, \Omega^*) \text{ is a minimizer of (12) with data } \mathbf{Z}\},$$

*where $\mathbf{Z}$ is the standardized version of $\mathbf{X}$. Then, for all sufficiently small $\lambda, \delta \ge 0$ and all $n$, we have $\mathcal{G}(\mathcal{O}_{n,\lambda,\delta}(\mathbf{X})) = \mathcal{G}(\mathcal{O}_{n,\lambda,\delta}(\mathbf{Z}))$. Moreover, for all sufficiently small $\lambda, \delta > 0$ we have*

$$P\left[\mathcal{G}(\mathcal{O}_{n,\lambda,\delta}(\mathbf{X})) = \mathcal{G}(\mathcal{O}_{n,\lambda,\delta}(\mathbf{Z})) = \mathcal{G}(\mathcal{E}_{\min}(\Theta_f(B^0, \Omega^0)))\right] \to 1 \quad \text{as } n \to \infty.$$

Thus, *even on finite samples*, the set of DAG structures $\mathcal{G}(\mathcal{O}_{n,\lambda,\delta}(\mathbf{X}))$ derived from the raw (unstandardized) data $\mathbf{X}$ will always be the same as $\mathcal{G}(\mathcal{O}_{n,\lambda,\delta}(\mathbf{Z}))$, which is derived from standardized data $\mathbf{Z}$. As a result, standardizing Gaussian data does not affect the recovered DAG structure if the optimization problem (12) can be solved exactly.

**Remark 2.** *Theorem 3 applies to* global *optimization of the objective* (12). *Of course, in practice, algorithms can get stuck in local optima, but the* global *solutions (even for finite samples $n$) will always be scale invariant.*

# 5 Nonconvex regularized log-likelihood for general models

The results in the previous section are not specific to Gaussian models, although this helps with interpretability in a familiar setting. We now extend these results from linear Gaussian SEMs to more general SEMs. Here, we assume that $X$ follows model (2) and the induced distribution is denoted by $P(X; \psi^0, \xi^0)$. Let us define the equivalence class $\mathcal{E}(\psi^0, \xi^0)$,

$$\mathcal{E}(\psi^0, \xi^0) = \{(\psi, \xi) : P(x; \psi, \xi) = P(x; \psi^0, \xi^0), \forall x \in \mathbb{R}^p\}.$$

That is, $\mathcal{E}(\psi^0, \xi^0)$ is a set of pairs $(\psi, \xi)$ that induce the same distribution $P(X; \psi^0, \xi^0)$. As a result, any pair $(\psi, \xi)$ within this equivalence class will be a minimizer of the NLL $\ell(\psi, \xi)$. Analogously to Definition 2, we can also define the collection of minimal elements in the equivalence class $\mathcal{E}(\psi^0, \xi^0)$.

**Definition 3.** $(\psi, \xi)$ *is called a minimal-edge I-map in the equivalence class* $\mathcal{E}(\psi^0, \xi^0)$ *if* $s_{B(\psi)} \leq s_{B(\widetilde{\psi})}, \forall (\widetilde{\psi}, \widetilde{\xi}) \in \mathcal{E}(\psi^0, \xi^0)$. *We further define*

$$\mathcal{E}_{\min}(\psi^0, \xi^0) = \{(\psi, \xi) : (\psi, \xi) \text{ is minimal-edge I-map}, (\psi, \xi) \in \mathcal{E}(\psi^0, \xi^0)\}.$$

Here, it is crucial that our concept of minimality concerns $s_{B(\psi)}$, which is the number of nonzero entries in the weighted adjacency matrix $B(\psi)$, rather than the number of nonzero entries in the parameter $\psi$ itself. Therefore, $s_{B(\psi)}$ essentially counts the number of edges in the adjacency matrix.

**Assumption A.** *(1)* $|\mathcal{E}(\psi^0, \xi^0)|$ *is finite. (2)* $B(\psi)$ *is L-Lipschitz w.r.t.* $\psi$, *i.e.* $\frac{\|B(\psi_1) - B(\psi_2)\|_2}{\|\psi_1 - \psi_2\|_2} \leq L$.

**Assumption B.** *For any* $\alpha$ *such that* $\ell(\psi^0, \xi^0) < \alpha$, *the level set* $\{(\psi, \xi) : \ell(\psi, \xi) \leq \alpha\}$ *is bounded, where* $\ell(\psi, \xi)$ *is the expected NLL defined in (5).*

Assumption A(1) is relatively mild; it requires that the equivalence class contains only finitely many points. This assumption is satisfied by Gaussian models, generalized linear models with continuous output [66], binary output [74, 13], and most exponential families. It is also obviously satisfied by any identifiable model since $|\mathcal{E}(\psi^0, \xi^0)| = 1$. Assumption A(2) is a mild continuity requirement on $B(\psi)$. Assumption B simply guarantees that the optimization problem has a minimizer, and is standard [7]. More discussions about the assumptions are included in Appendix C.7. Without this type of assumption, score-based learning is not even well-defined.

Similar in spirit to Theorem 1, we can show that by combining the NLL with quasi-MCP for appropriate $\lambda, \delta$, solving the following problem, we recover elements of $\mathcal{E}_{\min}(\psi^0, \xi^0)$:

$$\min_{\psi \in \Psi, \xi \in \Xi} \ell_n(\psi, \xi) + p_{\lambda, \delta}(B(\psi)) \quad \text{subject to} \quad h(B(\psi)) = 0, \tag{14}$$

where $p_{\lambda, \delta}(\cdot)$ is quasi-MCP defined in (10). Next, define its set of global minimizers.

$$\mathcal{O}_{n, \lambda, \delta} = \{(\psi^*, \xi^*) : (\psi^*, \xi^*) \text{ is minimizer of (14)}\}.$$

**Theorem 4.** *Let* $X$ *follow model* (2) *with parameters* $(\psi^0, \xi^0)$ *and let* $\mathbf{X}$ *be* $n$ *i.i.d. samples from* $P(X; \psi^0, \xi^0)$. *Under Assumptions A-B, for all sufficiently small* $\lambda, \delta > 0$ *(independent of* $n$), *it holds that* $P(\mathcal{O}_{n, \lambda, \delta} = \mathcal{E}_{\min}(\psi^0, \xi^0)) \to 1$ *as* $n \to \infty$.

**Theorem 5.** *Under the setting in Theorem 4 and assuming that* $P(X; \xi^0, \psi^0)$ *is faithful with respect to* $G^0 := G(B(\psi^0))$. *Then, for all sufficiently small* $\lambda, \delta > 0$ *(independent of* $n$), *it holds that* $P(\mathcal{O}_{n, \lambda, \delta} = \mathcal{M}(G^0)) \to 1$ *as* $n \to \infty$.

# 6 Experiments

To solve (12) and (14), we employ the augmented Lagrangian algorithm [5] from NOTEARS [73, 74], modifying their least squares score with $\ell_1$ penalty into the log-likelihood with MCP (10). We compare our approach to relevant baselines, e.g. NOTEARS [73], GOLEM [41], DAGMA [4], VarSort [55], FGES [52] and PC [59]. For our variation of NOTEARS that employs a score function based on the NLL with MCP, we name it as LOGLL-NOTEARS. The suffixes 'POPULATION' and 'SAMPLE' denote the use of the population and sample covariance matrix, respectively. Full details of the experiments are given in Appendix D. Open-source code implementing these methods is available at https://github.com/duntrain/dagrad.

Our primary empirical results are shown in Figures 1 and 2. We use the structural Hamming distance (SHD) as the main metric to evaluate the difference between the estimated graph and the ground truth graph. Lower SHD values indicate better estimation accuracy. Given that the model specified in (6) is nonidentifiable, we compare the CPDAGs of the estimated graph and the ground truth graph.

In Figure 1(a), we observe that using the NLL+MCP achieves the best performance for the different types of graphs and ranks second best for sparse graphs {ER1, SF1}. In Figure 1(b), standardizing **X** significantly impacts the performance of GOLEM, NOTEARS, and DAGMA; the SHD values are not any better than an empty graph, exactly as predicted by prior theory. The performance of LOGLL-NOTEARS-SAMPLE and LOGLL-NOTEARS-POPULATION are also affected by standardization, but these methods remain robust and continue to make meaningful discoveries. It is important to note that this observation does not contradict our Lemma 2. The challenges arise because solving the optimization problems (12) and (14) to find global solutions becomes inherently difficult as $p$ increases.To verify the scale invariance property in Theorem 3, we also conduct experiments on small graphs and include exact method that solve (12) and (14) to global optimal, see Figure 5.

In Figure 2, we replicate the Figure 1 in [55], providing a more direct comparison between various methods applied to raw data (**X**) and standardized data (**X** standardized). We include VarSort (referred to as sortnregress in [55]) as a baseline. Notably, for smaller graphs ($p = 10$), both LOGLL(-NOTEARS)-SAMPLE and LOGLL(-NOTEARS)-POPULATION exhibit the scale-invariant property alongside PC and FGES, in alignment with Lemma 2. This contrasts sharply with other methods, which completely deteriorate. For larger graphs ($p = 50$), standardizing the data mildly degrades the performance of LOGLL(-NOTEARS)-SAMPLE and LOGLL(-NOTEARS)-POPULATION. This can be attributed to the increased complexity of optimization as the size of the graph grows.

In Figure 3, we use a concrete toy example to investigate two key factors in the implementation: (1) the impact of random initialization, and (2) the upper limit for $\delta_0$ that can be applied according to Theorem 1. We generate $10^5$ initializations $B_{\text{random}}$ with weight for each edge uniformly sampled within $[-5, 5]$, and perform optimization using LOGLL-NOTEARS starting from these points. The "maximal $\delta$" is the theoretical maximum $\delta_0$ that ensures the validity of Theorem 1. We computed the SHD and the distances between the estimated $\mathcal{M}(B_{\text{est}})$ and $\mathcal{M}(B^0)$. The red line in Figure 3 represents the average SHD and distances. The distribution of these $10^5$ estimated SHD and distances is visualized using dots of varying sizes, where larger dots indicate a higher frequency of points. In some cases where SHD takes a value of $-1$, this value is used to indicate that the estimated $B_{\text{est}}$ does not form a valid DAG, which is an artifact of thresholding and affects $< 0.5\%$ of models. For the remaining models, the optimization (12) can typically be solved very close to a globally optimal, and according to Theorem 2, the SHD should ideally be zero, which is consistent with the figure.

Our results are not limited to the linear model with Gaussian noise. In Appendix E.3, we provide additional experiments on a logistic model (binary $X_j$) and neural networks. Further details on the experimental settings and additional experiments can be found in Appendix D and E.

# 7   Conclusion

Continuous score-based structure learning is a relative newcomer to the literature on causal structure learning, which goes back several decades. It has attracted significant attention due to its simplicity and generality, however, its theoretical properties are often misunderstood. We have sought to fill in this gap by studying its statistical aspects (to complement ongoing computational studies, e.g. [41, 65, 4, 13, 14, 43]). To this end, we proposed a fully differentiable score function for structure learning, composed of log-likelihood and quasi-MCP. We demonstrated that the global solution corresponds to the sparsest DAG structure that is Markov to the data distribution. Under mild assumptions, we conclude that all optimal solutions are the sparsest within the same Markov equivalence class. Additionally, the proposed score is scale-invariant, producing the same structure regardless of the data scale under the linear Gaussian model. Experimental results validate our theory, showing that our score provides better and more robust structure recovery compared to other scores.

We hope that this work stimulates further statistical inquiry into the properties of CSL. For example, we have focused on parametric models, and left extensions to nonparametric models to future work. Certain assumptions such as the finiteness of the equivalence class and the boundedness of the level set of the log-likelihood become more interesting in this regime. We have mentioned already that extensions to richer data types including interventions is an important direction. It would be of great

interest to explore ways to relax our assumptions to expand our statistical understanding of CSL in broader scenarios.

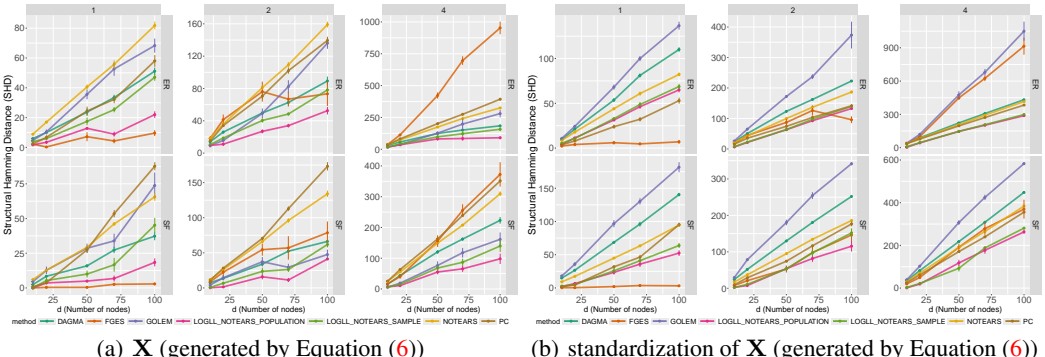

(a) **X** (generated by Equation (6))

(b) standardization of **X** (generated by Equation (6))

Figure 1: Results in terms of SHD between MECs of estimated graph and ground truth. Lower is better. Column: $k = \{1, 2, 4\}$. Row: random graph types. {ER,SF}-$k$ = {Scale-Free,Erds-Rényi } graphs with $kd$ expected edges. Here $p = \{10, 20, 50, 70, 100\}, n = 1000$.

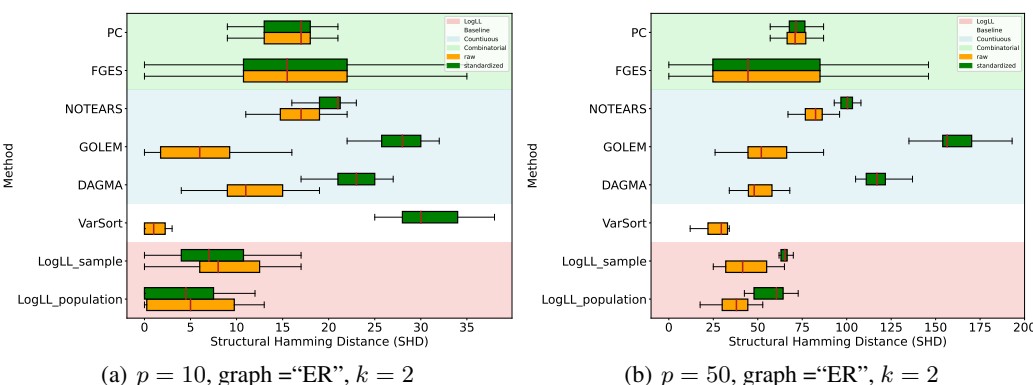

(a) $p = 10$, graph ="ER", $k = 2$

(b) $p = 50$, graph ="ER", $k = 2$

Figure 2: Comparison of raw (orange) vs. standardized (green) data. SHD (lower is better) between Markov equivalence classes (MEC) of recovered and ground truth graphs for ER-2 graphs with 10 (left) or 50 (right) nodes. In (b), SHD for VarSort with standardized data is omitted due to its average exceeding 300.

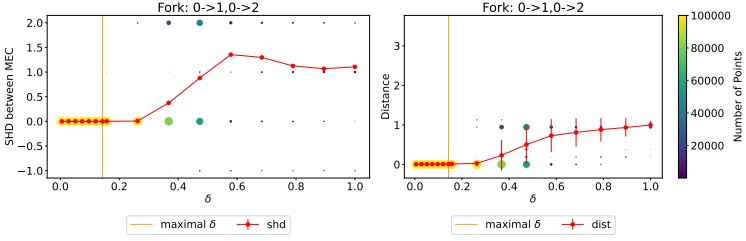

Figure 3: Graph: fork structure $X_0 \rightarrow X_1$ and $X_0 \rightarrow X_2$. For $0 < \delta < \delta_0$, the estimated $(B_{\text{est}}, \Omega_{\text{est}}) \in \mathcal{E}_{\min}(\Theta^0)$ because SHD and distance are close to 0.

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

# SUPPLEMENTARY MATERIAL
## Markov Equivalence and Consistency in
## Differentiable Structure Learning

## A   Preliminary Technical Results

In this appendix, we include various technical results used to prove the main theorems of the paper. Proofs can be found in Appendix B.

The following corollary supports Remark 1. In the main paper, we use quasi-MCP (10) as a penalty in the optimization problems (12) and (14) for simplicity. However, similar conclusions hold when MCP or SCAD is used as the penalty term.

**Corollary 1** (MCP/SCAD). *Under the same setting as Theorem 1. Let optimal solutions collection be*

$$\mathcal{O}_{n,\lambda,a} = \{(B^*, \Omega^*) : (B^*, \Omega^*) \text{ is minimizer of (12) with } p_{\lambda,\delta}(t) \text{ replaced by } p_{\lambda,a}^{MCP}(t) \text{ or } p_{\lambda,a}^{SCAD}(t)\}$$

*Then, for all sufficiently small $\lambda, a > 0$ (independent of $n$), it holds that $\mathcal{O}_{n,\lambda,a} = \mathcal{E}_{\min}(\Theta^0)$ as $n \to \infty$, where MCP $p_{\lambda,a}^{MCP}(\cdot)$ and SCAD $p_{\lambda,a}^{SCAD}(\cdot)$ are defined in Appendix C.5.*

The following lemma is a generalization of Lemma 1. Even for the general model, under the faithfulness assumption, all elements in the minimal equivalence class $\mathcal{E}_{\min}(\psi^0, \xi^0)$ belong to the same Markov equivalence class, as is the case in the general linear Gaussian model (6).

**Lemma 3.** *Consider that $X$ is generated by (2) with $(\psi^0, \xi^0)$. Assume that $P(X; \xi^0, \psi^0)$ is faithful to $G^0 := G(B(\psi^0))$. Then*

$$\mathcal{M}(G^0) = \mathcal{G}(\mathcal{E}_{\min}(\psi^0, \xi^0)) = \{G(B(\psi)) : (\psi, \xi) \in \mathcal{E}_{\min}(\psi^0, \xi^0)\}$$

*where $B(\psi)$ is the adjacency matrix implied by the parameterization $(\psi, \xi)$, see (3). $\mathcal{M}(G^0)$ is the Markov equivalence class of $G^0$, see Definition 1.*

Under the Sparsest Markov representation assumption, all elements in the minimal equivalence class are also in the same Markov equivalence class. It is important to note that the faithfulness assumption is stronger than the Sparsest Markov representation assumption. Specifically, if $P$ is faithful with respect to $G$, then the pair $(G, P)$ satisfies the Sparsest Markov representation assumption.

**Lemma 4.** *If a pair $(G, P(X; B, \Omega))$ satisfies Sparsest Markov representation (SMR) (see Definition 4), then $\mathcal{M}(G) = \mathcal{G}(\mathcal{E}_{\min}(\Theta)) = \{G(B) : (B, \Omega) \in \mathcal{E}_{\min}(\Theta)\}$ where $\mathcal{M}(G)$ is Markov equivalence class of $G$ (see Definition 1).*

The following lemma provides the formulation for the standardization of $X$, along with its covariance and precision matrices.

**Lemma 5** (standardization). *Let $X \sim \mathcal{N}(0, \Sigma)$, $\sigma_i^2 := \mathrm{Var}(X_i)$ and $D := \mathrm{diag}(\sigma_1, \dots, \sigma_p)$. Then the standardization of $X$, corresponding covariance matrix and precision matrix can be expressed as*

$$X_{\mathrm{std}} := D^{-1}(X - \mathbb{E}X), \quad \mathrm{Cov}(X_{\mathrm{std}}) = D^{-1}\Sigma D^{-1} \quad [\mathrm{Cov}(X_{\mathrm{std}})]^{-1} = D\Theta D$$

The following lemma establishes an useful identity that holds for any adjacency matrix of a DAG, which is used in the derivation of the log-likelihood function for the model in Equation (6).

**Lemma 6.** *If $B$ is adjacency matrix of a DAG, then $\log\det(I - B) = 0$.*

The following lemma provides a condition under which the optimization problem (12) is well-defined, ensuring that $\ell(B, \Omega) > -\infty$ for any $(B, \Omega)$.

**Lemma 7.** *For any $(B, \Omega)$, if $\Omega > 0$, then $\Sigma := \Sigma_f(B, \Omega)$ is positive definite. Moreover, if $X$ is generated by Equation (6) with $(B^0, \Omega^0)$, then $\ell(B, \Omega) > -\infty$ for any $(B, \Omega)$.*

The following lemma is used in the proof of Theorem 1. It justifies that the loss of every element in $A_3$ is strictly greater than the loss of the ground truth, i.e., $(B^0, \Omega^0)$.

**Lemma 8.** *Under the same setting and notation as in the proof of Theorem 1, see Section B.1. If for any $(\bar{B}, \bar{\Omega}) \in \mathcal{E}(\Theta^0)$, it holds that $dist(\bar{B}, A_3) > 0$, there exists $\alpha > 0$ such that $\ell(B, \Omega) - \ell(B^0, \Omega^0) > \alpha$ for all $(B, \Omega) \in A_3$.*

# B  Detailed Proofs

## B.1  Proof of Theorem 1

*Proof.* It suffices to consider the population case, i.e., $\ell_n(B, \Omega)$ is replaced by its population counterpart $\ell(B, \Omega)$. By Lemma 7, we have

$$\ell(B, \Omega) > -\infty$$

Also, $p_{\lambda,\delta}(B) \geq 0$ for any $B$. Consequently, optimization problem (12) is well-defined.

By convention, we assume that $(B^0, \Omega^0) \in \mathcal{E}_{\min}(\Theta^0)$. Now, consider the case where $p_{\lambda,\delta}(B^0) = 0$, which is equivalent to $B^0 = 0$, since $\ell(B, \Omega) \geq \ell(B^0, \Omega^0)$ and $p_{\lambda,\delta}(B) \geq p_{\lambda,\delta}(B^0)$ for any $B$. Therefore, for all $\lambda > 0$ and $\delta > 0$, $B^0$ is the unique optimal solution to optimization problem (12), proving the conclusion.

In the subsequent proof, we assume that $|\mathcal{E}_{\min}(\Theta^0)| = 1$, that is, $\mathcal{E}_{\min}(\Theta^0) = \{(B^0, \Omega^0)\}$. This assumption simplifies the proof because any element of $\mathcal{E}_{\min}(\Theta^0)$ is indistinguishable based on the value of $\ell(B, \Omega)$ and the penalty for the chosen $(\delta, \lambda)$, as shown below. Our goal is to identify one element via optimization problem (12), which significantly simplifies the argument.

First, let us define

$$\delta_0 = \frac{\tau}{1 + \Delta} \qquad \text{where } \tau := \min_{(B,\Omega) \in \mathcal{E}(\Theta^0)} \min_{\{(i,j) \mid B_{ij} \neq 0\}} |B_{ij}| \stackrel{(a)}{=} \min_{\pi} \min_{\{(i,j) \mid [\widetilde{B}^0(\pi)]_{ij} \neq 0\}} \left| [\widetilde{B}^0(\pi)]_{ij} \right|$$

with any $\Delta > 0$. $(a)$ is due to the fact that each element in equivalence class $\mathcal{E}(\Theta^0)$ is one-to-one associated with $\widetilde{B}^0(\pi)$, see Section C.2 or [2] for detailed discussion. Then, for any $\lambda > 0$ and $0 < \delta < \delta_0$, consider the set $A_1 = \{(B, \Omega) \mid p_{\lambda,\delta}(B) = p_{\lambda,\delta}(B^0)\}$. For any $(B, \Omega) \in A_1$, we have $(B, \Omega) \notin \mathcal{E}(\Theta^0) \setminus \{(B^0, \Omega^0)\}$. This follows from the fact that

$$p_{\lambda,\delta}(B) = \frac{\lambda\delta}{2} s_B > \frac{\lambda\delta}{2} s_{B^0} = p_{\lambda,\delta}(B^0) \qquad \forall (B, \Omega) \in \mathcal{E}(\Theta^0) \setminus \{(B^0, \Omega^0)\}.$$

As a consequence, this implies that $\ell(B^0, \Omega^0) < \ell(B, \Omega), \forall (B, \Omega) \in A_1$. Therefore,

$$\ell(B^0, \Omega^0) + p_{\lambda,\delta}(B^0) < \ell(B, \Omega) + p_{\lambda,\delta}(B) \qquad \forall (B, \Omega) \in A_1.$$

Next, we define $A_2 = \{(B, \Omega) \mid p_{\lambda,\delta}(B) > p_{\lambda,\delta}(B^0)\}$. Since $\ell(B^0, \Omega^0) \leq \ell(B, \Omega)$, it follows that for all $(B, \Omega) \in A_2$, the following inequality holds:

$$\ell(B^0, \Omega^0) + p_{\lambda,\delta}(B^0) < \ell(B, \Omega) + p_{\lambda,\delta}(B) \qquad \forall (B, \Omega) \in A_2.$$

Therefore, we need to examine the set $A_3 = \{(B, \Omega) \mid p_{\lambda,\delta}(B) < p_{\lambda,\delta}(B^0)\}$. For $(B^0, \Omega^0)$ to achieve the minimum value of the score function, it is crucial that the following condition is satisfied:

$$\ell(B^0, \Omega^0) + p_{\lambda,\delta}(B^0) < \ell(B, \Omega) + p_{\lambda,\delta}(B) \qquad \forall (B, \Omega) \in A_3.$$

This condition guarantees that the ground truth parameters $(B^0, \Omega^0)$ correspond to the optimal solution by comparing their score with any other parameters in the subset $A_3$.

It is important to note that $p_{\lambda,\delta}(t) = \lambda p_{1,\delta}(t), \forall t$. Thus, a necessary and sufficient condition for this to hold is:

$$\lambda < \min_{(B,\Omega) \in A_3} \frac{\ell(B, \Omega) - \ell(B^0, \Omega^0)}{p_{1,\delta}(B^0) - p_{1,\delta}(B)}.$$

Note that for all $(B, \Omega) \in A_3$, we have $p_{1,\delta}(B^0) - p_{1,\delta}(B) \leq \frac{\delta}{2} s_0$, with equality achieved when $B = 0$. Therefore, the denominator on the RHS cannot be arbitrarily large. Moreover, since $(B, \Omega) \in A_3$, it follows that $(B, \Omega) \notin \mathcal{E}(\Theta^0)$, as $A_3 \cap \mathcal{E}(\Theta^0) = \emptyset$.

We define the distance from $\bar{B}$ to the set $A_3$ as:

$$\mathrm{dist}(\bar{B}, A_3) = \inf_{(B,\Omega)\in A_3} \|B - \bar{B}\|_2.$$

For all $(\bar{B}, \bar{\Omega}) \in \mathcal{E}(\Theta^0)$, it turns out that $\mathrm{dist}(\bar{B}, A_3)$ must be positive due to the design of $\delta_0$, giving:

$$\mathrm{dist}(\bar{B}, A_3) > \min_{(B,\Omega)\in\mathcal{E}(\Theta^0)} \min_{\{(i,j)|B_{ij}\neq 0\}} |B_{ij}| - \delta_0$$

$$= \tau - \frac{\tau}{1 + \Delta} = \frac{\Delta}{1 + \Delta}\tau > 0.$$

By Lemma 8, there exists some $\alpha > 0$ such that $\ell(B, \Omega) - \ell(B^0, \Omega^0) > \alpha$ for all $(B, \Omega) \in A_3$. Consequently, we have:

$$\inf_{(B,\Omega)\in A_3} \frac{\ell(B, \Omega) - \ell(B^0, \Omega^0)}{p_{1,\delta}(B^0) - p_{1,\delta}(B)} > 0.$$

Thus, we can define

$$\lambda_0 = \inf_{(B,\Omega)\in A_3} \frac{\ell(B, \Omega) - \ell(B^0, \Omega^0)}{p_{1,\delta}(B^0) - p_{1,\delta}(B)} > 0.$$

In summary, for all $0 < \lambda < \lambda_0$ and $0 < \delta < \delta_0$, for any $(\widehat{B}, \widehat{\Omega}) \in \mathcal{E}_{\min}(\Theta^0)$, and for all $(B, \Omega) \notin \mathcal{E}_{\min}(\Theta^0)$, the following holds:

$$\ell(\widehat{B}, \widehat{\Omega}) + p_{\lambda,\delta}(\widehat{B}) < \ell(B, \Omega) + p_{\lambda,\delta}(B).$$

This concludes the proof. $\qquad \square$

## B.2   Proof of Theorem 2

*Proof.* Here, $\Theta_f(B^0, \Omega^0) = \Theta^0$. From Theorem 1, we know that when $n \to \infty$

$$\mathcal{O}_{n,\lambda,\delta} = \mathcal{E}_{\min}(\Theta^0).$$

Given the additional assumption that $p(X)$ is faithful with respect to $G^0 := G(B^0)$, by Lemma 1, we have

$$\mathcal{M}(G^0) = \mathcal{E}_{\min}(\Theta^0) = \{G(B) : (B, \Omega) \in \mathcal{E}_{\min}(\Theta^0)\}.$$

Note that

$$\{G(B) : (B, \Omega) \in \mathcal{E}_{\min}(\Theta^0)\} = \{G(B) : (B, \Omega) \in \mathcal{O}_{n,\lambda,\delta}\}.$$

Thus, we conclude that

$$\mathcal{M}(G^0) = \mathcal{O}_{n,\lambda,\delta}. \qquad \text{as } n \to \infty$$

This completes the proof. $\qquad \square$

## B.3   Proof of Theorem 3

*Proof.* In this proof, we use the notation introduced in Section C.6. Note that when $\mathbf{X}$ is used in (12), we essentially compute the sample covariance matrix $\widehat{\Sigma}$ based on $\mathbf{X}$ as follows:

$$\widehat{\Sigma} = \frac{1}{n} \left[ \mathbf{X} - \mathbf{1}_n \cdot (\widehat{\mu}_1, \ldots, \widehat{\mu}_p) \right]^\top \left[ \mathbf{X} - \mathbf{1}_n \cdot (\widehat{\mu}_1, \ldots, \widehat{\mu}_p) \right]$$

and plug it into the negative sample log-likelihood function. The same procedure applies to $\mathbf{Z}$:

$$\widehat{\Sigma}_{\mathrm{std}} = \frac{1}{n} D^{-1} \left[ \mathbf{X} - \mathbf{1}_n(\widehat{\mu}_1, \ldots, \widehat{\mu}_p) \right]^\top \left[ \mathbf{X} - \mathbf{1}_n(\widehat{\mu}_1, \ldots, \widehat{\mu}_p) \right] D^{-1}$$

$$= D^{-1} \widehat{\Sigma} D^{-1}.$$

Denote $\widehat{\Theta} = (\widehat{\Sigma})^{-1}$ and $\widehat{\Theta}_{\text{std}} = (\widehat{\Sigma}_{\text{std}})^{-1} = D\widehat{\Theta}D$. For $\widehat{\Theta}$, by applying Theorem 1, there exist $\lambda_0^{\text{raw},n} > 0$ and $\delta_0^{\text{raw},n} > 0$ such that for any $0 < \lambda < \lambda_0^{\text{raw},n}$ and $0 < \delta < \delta_0^{\text{raw},n}$, we have $\mathcal{O}_{n,\lambda,\delta}(\mathbf{X}) = \mathcal{E}_{\min}(\widehat{\Theta})$. For $D\widehat{\Theta}D$, we apply Theorem 1 again, and there exist $\lambda_0^{\text{std},n} > 0$ and $\delta_0^{\text{std},n} > 0$ such that for any $0 < \lambda < \lambda_0^{\text{std},n}$ and $0 < \delta < \delta_0^{\text{std},n}$, we have $\mathcal{O}_{n,\lambda,\delta}(\mathbf{Z}) = \mathcal{E}_{\min}(D\widehat{\Theta}D)$.

We can select $\delta_0 = \min\{\delta_0^{\text{raw},n}, \delta_0^{\text{std},n}\}$ and $\lambda_0 = \min\{\lambda_0^{\text{raw},n}, \lambda_0^{\text{std},n}\}$ in optimization (12) to ensure that $\mathcal{O}_{n,\lambda,\delta}(\mathbf{X}) = \mathcal{E}_{\min}(\widehat{\Theta})$ and $\mathcal{O}_{n,\lambda,\delta}(\mathbf{Z}) = \mathcal{E}_{\min}(D\widehat{\Theta}D)$ hold simultaneously.

By applying Lemma 2, we conclude that:

$$\mathcal{G}(\mathcal{O}_{n,\lambda,\delta}(\mathbf{X})) = \mathcal{G}(\mathcal{E}_{\min}(\widehat{\Theta})) = \mathcal{G}(\mathcal{E}_{\min}(D\widehat{\Theta}D)) = \mathcal{G}(\mathcal{O}_{n,\lambda,\delta}(\mathbf{Z})).$$

Furthermore, as $n \to \infty$, we have $\widehat{\Sigma} \to \Sigma$ and $\widehat{\Theta} \to \Theta$. Therefore, $\mathcal{E}_{\min}(\widehat{\Theta}) \to \mathcal{E}_{\min}(\Theta)$, and $\mathcal{E}_{\min}(\widehat{\Theta}_{\text{std}}) \to \mathcal{E}_{\min}(\Theta_{\text{std}})$ as $n \to \infty$. Thus,

$$\mathcal{G}(\mathcal{O}_{n,\lambda,\delta}(\mathbf{X})) = \mathcal{G}(\mathcal{E}_{\min}(\Theta)) = \mathcal{G}(\mathcal{O}_{n,\lambda,\delta}(\mathbf{Z})). \qquad \text{where } n \to \infty$$

Note that we use $n \to \infty$ to indicate we consider the result in population level. $\qquad\square$

## B.4  Proof of Theorem 4

*Proof.* This proof shares many similarities with the proof of Theorem 1. First, when $n \to \infty$, we consider the result at the population level. Thus, $\ell_n(\psi, \xi) \to \ell(\psi, \xi)$, and we will focus on $\ell(\psi, \xi)$ in the following. As a result, we only work with $\ell(\psi, \xi)$ instead of $\ell_n(\psi, \xi)$.

By convention, we can always assume that $(\psi^0, \xi^0) \in \mathcal{E}_{\min}(\psi^0, \xi^0)$.

Now, consider the case where $p_{\lambda,\delta}(B(\psi^0)) = 0$, which implies that $B(\psi^0) = 0$. Since $X \sim P(X, \psi^0, \xi^0)$, we have

$$\ell(\psi^0, \xi^0) \le \ell(\psi, \xi),$$

and thus

$$\ell(\psi^0, \xi^0) + p_{\lambda,\delta}(B(\psi^0)) \le \ell(\psi, \xi) + p_{\lambda,\delta}(B(\psi)) \qquad \forall(\psi, \xi) \in \Psi \times \Xi, \forall \lambda > 0, \forall \delta > 0.$$

Therefore, $(\psi^0, \xi^0)$ is the optimal solution to optimization (12).

As we iterate, we can assume that $|\mathcal{E}_{\min}(\psi^0, \xi^0)| = 1$, meaning $\mathcal{E}_{\min}(\psi^0, \xi^0) = \{(\psi^0, \xi^0)\}$. This assumption simplifies the proof because any element in $\mathcal{E}_{\min}(\psi^0, \xi^0)$ is indistinguishable based on the value of $\ell(\psi, \xi)$ and the penalty for the chosen parameters $(\delta, \lambda)$. Our goal is to find one element by solving the optimization problem (14), and this assumption simplifies the argument significantly.

First, we define

$$\delta_0 = \frac{\tau}{1 + \Delta} \qquad \tau := \min_{(\psi, \xi) \in \mathcal{E}(\psi^0, \xi^0)} \min_{\{(i,j) : B(\psi)_{ij} \neq 0\}} |B(\psi)|_{ij}.$$

It is important to note that, under Assumption A (1), since $|\mathcal{E}(\psi^0, \xi^0)|$ is finite, we have $\tau > 0$.

Then, for any $\lambda > 0$ and $0 < \delta < \delta_0$, consider the set

$$A_1 = \{(\psi, \xi) \mid p_{\lambda,\delta}(B(\psi)) = p_{\lambda,\delta}(B(\psi^0))\}.$$

It is clear that for any $(\psi, \xi) \in A_1$, we must have $(\psi, \xi) \notin \mathcal{E}(\psi^0, \xi^0)$, since for any $(\psi, \xi) \in \mathcal{E}(\psi^0, \xi^0)$, we have $p_{\lambda,\delta}(B(\psi)) > p_{\lambda,\delta}(B(\psi^0))$. As a result,

$$\ell(\psi^0, \xi^0) < \ell(\psi, \xi) \qquad \forall(\psi, \xi) \in A_1.$$

Therefore,

$$\ell(\psi^0, \xi^0) + p_{\lambda,\delta}(B(\psi^0)) < \ell(\psi, \xi) + p_{\lambda,\delta}(B(\psi)) \qquad \forall(\psi, \xi) \in A_1.$$

Next, we consider the set $A_2 = \{(\psi, \xi) \mid p_{\lambda,\delta}(B(\psi)) > p_{\lambda,\delta}(B(\psi^0))\}$, and we know that $\ell(\psi^0, \xi^0) \le \ell(\psi, \xi)$. Therefore,

$$\ell(\psi^0, \xi^0) + p_{\lambda,\delta}(B(\psi^0)) < \ell(\psi, \xi) + p_{\lambda,\delta}(B(\psi)) \qquad \forall(\psi, \xi) \in A_2.$$

Consequently, we need to check $A_3 = \{(\psi, \xi) \mid p_{\lambda,\delta}(B(\psi)) < p_{\lambda,\delta}(B(\psi^0))\}$. For $(\psi^0, \xi^0)$ to be the minimizer of (14), we require that the following condition holds:

$$\ell(\psi^0, \xi^0) + p_{\lambda,\delta}(B(\psi^0)) < \ell(\psi, \xi) + p_{\lambda,\delta}(B(\psi)) \qquad \forall (\psi, \xi) \in A_3.$$

This condition ensures that the ground truth parameters $(\psi^0, \xi^0)$ correspond to the optimal solution by comparing their score with that of any other parameters in the subset $A_3$.

It is also worth noting that $p_{\lambda,\delta}(t) = \lambda p_{1,\delta}(t)$ for all $t$. Therefore, a necessary and sufficient condition for this to hold is:

$$\lambda < \inf_{(\psi,\xi) \in A_3} \frac{\ell(\psi, \xi) - \ell(\psi^0, \xi^0)}{p_{1,\delta}(B(\psi^0)) - p_{1,\delta}(B(\psi))}.$$

Note that for $(\psi, \xi) \in A_3$, we have $p_{1,\delta}(B(\psi^0)) - p_{1,\delta}(B(\psi)) \leq \frac{\delta}{2} s_{B(\psi^0)}$. Therefore, the denominator on the RHS cannot be arbitrarily large. Moreover, for any $0 < \delta < \delta_0$, the following holds:

$$\frac{\Delta}{1 + \Delta} \tau \leq \|B(\psi^0) - B(\psi)\|_2 \leq L \|\psi^0 - \psi\|_2 \qquad \forall (\psi, \xi) \in A_3.$$

The second inequality follows from Assumption A (b). As a consequence,

$$\|(\psi, \xi) - (\psi^0, \xi^0)\|_2 \geq \|\psi^0 - \psi\|_2 \geq \frac{\tau \Delta}{L(1 + \Delta)}.$$

Thus, we obtain

$$A_3 \subseteq \{(\psi, \xi) \mid \|(\psi, \xi) - (\psi^0, \xi^0)\|_2 \geq \frac{\Delta \tau}{L(1 + \Delta)}\} = A_4.$$

First, note that $A_3$ is a nonempty set. Otherwise, the conclusion would hold immediately. Let us select any $(\bar{\psi}, \bar{\xi}) \in A_3 \neq \emptyset$, and define $A_5 = \{(\psi, \xi) \mid \ell(\psi, \xi) \leq \ell(\bar{\psi}, \bar{\xi})\}$. Then,

$$\inf_{(\psi,\xi) \in A_3} \ell(\psi, \xi) = \inf_{(\psi,\xi) \in A_3 \cap A_5} \ell(\psi, \xi) \geq \inf_{(\psi,\xi) \in A_4 \cap A_5} \ell(\psi, \xi).$$

It is important to note that $A_4 \cap A_5$ is nonempty, since $(\bar{\psi}, \bar{\xi}) \in A_4 \cap A_5$. By Assumption B and the properties of $\ell(\psi, \xi)$, we know that $A_5$ is a bounded and closed set, and $A_4$ is a closed set. Consequently, $A_4 \cap A_5$ is compact. Furthermore, for all $(\psi, \xi) \in \mathcal{E}(\psi^0, \xi^0)$, we have $(\psi, \xi) \notin A_4 \cap A_5$. All of this leads to the following conclusion:

$$\inf_{(\psi,\xi) \in A_3} \ell(\psi, \xi) \geq \min_{(\psi,\xi) \in A_4 \cap A_5} \ell(\psi, \xi) = \inf_{(\psi,\xi) \in A_4 \cap A_5} \ell(\psi, \xi) > \ell(\psi^0, \xi^0).$$

As a result, we define $\lambda_0$ as follows:

$$\lambda_0 = \inf_{(\psi,\xi) \in A_3} \frac{\ell(\psi, \xi) - \ell(\psi^0, \xi^0)}{p_{1,\delta}(B(\psi^0)) - p_{1,\delta}(B(\psi))} > 0. \qquad \square$$

## B.5 Proof of Theorem 5

*Proof.* The proof is combination of Lemma 3 and Theorem 4, similar to Proof of Theorem 2. $\square$

## B.6 Proof of Lemma 1

Before proving the result, we introduce the definitions of the Sparsest Markov representation assumption and restricted faithfulness, along with a few useful theorems.

**Definition 4** (Sparsest Markov representation [54])**.** *A pair $(G^0, P)$ satisfies the Sparsest Markov Representation (SMR) assumption if $(G^0, P)$ satisfies the Markov property and $|G| > |G^0|$ for every DAG $G$ such that $(G, P)$ satisfies the Markov property and $G \notin \mathcal{M}(G^0)$.*

In other words, the SMR assumption asserts that the true DAG $G^0$ is the (unique up to Markov equivalence) sparsest DAG satisfying the Markov property.

**Definition 5** (Restricted-faithfulness [53, 54]). *A distribution $P$ satisfies the restricted-faithfulness assumption with respect to a DAG $G$ if it is Markov to $G$ and following two conditions hold:*

- ***Adjacency-faithfulness:*** *for all $(j, k) \in E$ and all subsets $S \subset [p] \backslash \{j, k\}$ it holds that $X_j \not\perp\!\!\!\perp X_k \mid X_S$*

- ***Orientation-faithfulness:*** *for all triples $(j, k, l)$ with skeleton $j - l - k$ and all subsets $S \subset [p] \backslash \{j, k\}$ such that $j$ is d-connected to $k$ given $S$ it holds that $X_j \not\perp\!\!\!\perp X_k \mid X_S$*

**Theorem 6** ([53]). *If a distribution $P$ is faithful to $G$, then such distribution $P$ also satisfies the restricted-faithfulness assumption with respect to $G$.*

**Theorem 7** (Theorem 2.4 in [54]). *Let $(G, P)$ satisfy the Markov property. Then the restricted-faithfulness assumption implies the SMR assumption.*

*Proof.* First, by Theorem 6, the faithfulness assumption implies the restricted faithfulness assumption. Second, by Theorem 7, the restricted faithfulness assumption implies the Sparsest Markov representation assumption. Furthermore, note that for any $(B, \Omega) \in \mathcal{G}(\mathcal{E}_{\min}(\Theta^0))$, the distribution $P(X)$ is Markov to $G(B)$, since $(B, \Omega) \in \mathcal{G}(\mathcal{E}(\Theta^0))$. According to the definition of the Sparsest Markov representation assumption, all sparsest DAGs that satisfy the Markov property must belong to the same Markov equivalence class. In our case, this means $\mathcal{G}(\mathcal{E}_{\min}(\Theta^0)) = \mathcal{M}(G^0)$. $\qquad\square$

### B.7 Proof of Lemma 2

*Proof.* For $X \sim \mathcal{N}(0, \Sigma)$, let $\bar{\Theta} = D\Theta D$, and denote the inverse of $\bar{\Theta}$ as $\bar{\Sigma} = (\bar{\Theta})^{-1} = D^{-1}\Sigma D^{-1}$. It follows that $\bar{X} := D^{-1}X \sim \mathcal{N}(0, \bar{\Sigma})$. Now, consider the following least squares regression for $j \in \{1, \ldots, p\}$ and $S \subseteq \{1, \ldots, p\} \backslash \{j\}$. Let $\beta \in \mathbb{R}^{|S|}$. Then the following relationships hold:

$$\beta_{Sj} = \arg\min_\beta \mathbb{E}\|X_j - \beta^\top X_S\|_2^2 \Rightarrow \beta_{Sj} = \Sigma_{SS}^{-1}\Sigma_{Sj}$$

$$\bar{\beta}_{Sj} = \arg\min_\beta \mathbb{E}\|\bar{X}_j - \bar{\beta}^\top \bar{X}_S\|_2^2 \Rightarrow \bar{\beta}_{Sj} = \bar{\Sigma}_{SS}^{-1}\bar{\Sigma}_{Sj}$$

$$\bar{\Sigma}_{SS}^{-1} = ([D^{-1}\Sigma D^{-1}]_{SS})^{-1} = D_{SS}\Sigma_{SS}^{-1}D_{SS}$$

$$\bar{\Sigma}_{Sj} = [D^{-1}\Sigma D^{-1}]_{Sj} = D_{SS}^{-1}\Sigma_{Sj}D_{jj}^{-1}$$

$$\bar{\beta}_{Sj} = \bar{\Sigma}_{SS}^{-1}\bar{\Sigma}_{Sj} = D_{SS}\Sigma_{SS}^{-1}D_{SS}D_{SS}^{-1}\Sigma_{Sj}D_{jj}^{-1} = D_{SS}\beta_{Sj}D_{jj}^{-1}$$

As a consequence, $\text{supp}(\beta_{Sj}) = \text{supp}(\bar{\beta}_{Sj})$. Note that for all $(B, \Omega) \in \mathcal{E}(\Theta)$, we know from Section C.2 that there exists a $\pi \in \mathcal{P}$ such that $B = \widetilde{B}(\pi)$. Moreover, $B$ can be recovered by least squares regression using $X$ with its topological sort [2, 13] that is consistent with $\pi$. For such a $\pi$, we can find a pair $(\bar{B}, \bar{\Omega}) \in \mathcal{E}(D\Theta D)$, where $\bar{B}$ has the same topological sort as $\pi$, and it can be recovered by least squares regression on $\bar{X}$. We have shown that, for the same $S, j$, $\text{supp}(\beta_{Sj}) = \text{supp}(\bar{\beta}_{Sj})$. Therefore, $\text{supp}(B) = \text{supp}(\bar{B})$. $\qquad\square$

### B.8 Proof of Lemma 3

*Proof.* The proof is the same as Lemma 1. $\qquad\square$

### B.9 Proof Lemma 4

*Proof.* This follows directly from the definition of the Sparsest Markov Representation (SMR) assumption. Since for all $(B, \Omega) \in \mathcal{E}_{\min}(\Theta)$, $G(B)$ is Markovian to $P$ and $G(B)$ is the Sparsest, by Definition 2, all $G(B)$ must belong to the same Markov equivalence class by the definition of SMR. $\qquad\square$

### B.10 Proof of Lemma 5

*Proof.* $X_{\text{std}} = D^{-1}(X - \mathbb{E}X)$ is based on definition of standardization.

$$\text{Cov}(X_{\text{std}}) = \text{Cov}(D^{-1}(X - \mathbb{E}X)) = D^{-1}\text{Cov}((X - \mathbb{E}X))D^{-1} = D^{-1}\Sigma D^{-1}$$

$$[\text{Cov}(X_{\text{std}})]^{-1} = [D^{-1}\Sigma D^{-1}]^{-1} = D\Sigma^{-1}D = D\Theta D. \qquad\square$$

### B.11 Proof of Lemma 6

*Proof.* Detailed proof can be found in [41], Appendix Section D. ∎

### B.12 Proof of Lemma 7

*Proof.* From the definition of $\Sigma_f(B, \Omega)$:

$$\Sigma_f(B, \Omega) := (I - B)^{-\top}\Omega(I - B)^{-1} = (I - B)^{-\top}\Omega^{1/2}\Omega^{1/2}(I - B)^{-1},$$

where $\Omega^{1/2} = \mathrm{diag}(\omega_1, \ldots, \omega_p)$. It is clear that $\Sigma(B, \Omega)$ is positive semidefinite, as

$$x^\top\Sigma(B, \Omega)x = \|\Omega^{1/2}(I - B)^{-1}x\|_2^2 \geq 0, \qquad \forall x \in \mathbb{R}^p.$$

Next, we just need to show that $\Omega^{1/2}(I - B)^{-1}x \neq 0$ for all $x \neq 0$.

$$\Omega^{1/2}(I - B)^{-1}x \neq 0 \Leftrightarrow (I - B)^{-1}x \neq 0 \Leftrightarrow x \neq 0.$$

Here, $\omega_j^2 > 0$ for all $j$, so $\Omega^{1/2}$ is invertible. As $(I - B)$ is a full rank matrix, then $(I - B)^{-1}$ is also a full rank matrix, it indicates that $\Sigma(B, \Omega)$ is positive definite matrix.

Since $\Omega^0 > 0$, it follows that $\Sigma^0$ is positive definite. By Lemma 6, we have:

$$\begin{aligned}
\ell(B, \Omega) =& \frac{1}{2}\log\det\Omega - \log\det(I - B) + \frac{1}{2}\mathrm{Tr}(\Sigma^0\Theta(B, \Omega)) + \mathrm{const.} \\
=& \frac{1}{2}\log\det\Omega + \frac{1}{2}\mathrm{Tr}(\Sigma^0\Theta(B, \Omega)) + \mathrm{const.} \\
\geq& \ell(B, \Omega_f(B)) = \ell(B)
\end{aligned}$$

where $\Omega_f(B)$ and $\ell(B)$ are defined in Equations (15) and (17), respectively. The last inequality follows from Section C.1. Next, we need to prove that $\ell(B) > -\infty$.

$$\begin{aligned}
\ell(B) =& \frac{1}{2}\log\det\mathrm{diag}((I - B)^\top\Sigma^0(I - B)) + \mathrm{const.} \\
=& \frac{1}{2}\sum_{j=1}^p \log\mathbb{E}\|X_j - B_j^\top X\|_2^2 + \mathrm{const.} \\
=& \frac{1}{2}\sum_{j=1}^p \log\mathbb{E}\|(e_j - B_j)^\top X\|_2^2 + \mathrm{const.} \\
=& \frac{1}{2}\sum_{j=1}^p \log(e_j - B_j)^\top\Sigma^0(e_j - B_j) + \mathrm{const.} \\
\geq& \frac{1}{2}\sum_{j=1}^p \log\|(e_j - B_j)\|_2^2\Lambda_{\min}(\Sigma^0) + \mathrm{const.} \\
\geq& \frac{1}{2}\sum_{j=1}^p \log\Lambda_{\min}(\Sigma^0) > -\infty
\end{aligned}$$

Here, $e_j \in \mathbb{R}^p$ is a unit vector with the $j$-th position equal to 1 and all other positions being zero, and $\Lambda_{\min}(\Sigma^0)$ is the minimum eigenvalue of $\Sigma^0$. Since $\Sigma^0$ is positive definite, we have $\Lambda_{\min}(\Sigma^0) > 0$.

Because $B$ is the adjacency matrix of a DAG, it follows that $B_{jj} = 0$, which implies $\|e_j - B_j\| \geq \|e_j\| = 1$. As a result,

$$\ell(B, \Omega) \geq \ell(B) > -\infty. \qquad \Box$$

### B.13 Proof of Lemma 8

*Proof.* Note that for a fixed $B$, the corresponding optimal $\Omega_f(B) = \mathrm{diag}((I - B)^\top\Sigma^0(I - B))$ is the solution with respect to $\ell(B, \Omega)$. Therefore, without causing confusion, we take $\Omega_f(B) =$

$\mathrm{diag}((I - B)^\top \Sigma^0 (I - B))$ and consider the log-likelihood as a function of $B$ only, i.e., $\ell(B)$, for simpler representation. See Equation (17) in Section C.1 for details. It is clear that $0 \in A_3$, so we define $A_4 = \{B \mid \ell(B) \leq \ell(0)\}$. Note that $\ell(0)$ is finite.

$$\ell(0) \geq \ell(B) = \sum_{j=1}^{p} \log(e_j - B_j)^\top \Sigma^0 (e_j - B_j)$$

$$\geq \sum_{j=1}^{p} \log \|e_j - B_j\|^2 \Lambda_{\min}(\Sigma^0)$$

$$= \log \left[ (\Lambda_{\min}(\Sigma^0))^p \prod_{j=1}^{p} \|e_j - B_j\|^2 \right].$$

This indicates that

$$\prod_{j=1}^{p} \|e_j - B_j\|^2 \leq \frac{\exp(\ell(0))}{(\Lambda_{\min}(\Sigma^0))^p}$$

Moreover,

$$\|e_k - B_k\|^2 \leq \prod_{j=1}^{p} \|e_j - B_j\|^2 \leq \frac{\exp(\ell(0))}{\Lambda_{\min}^p} \qquad \forall k \in \{1, \ldots, p\}$$

This implies that $B_k$ must be bounded, and therefore every $B$ in $A_4$ is bounded. It is clear that $\arg\min_{B \in A_3} \ell(B) \in A_4$. Thus, we need to show that $\min_{B \in A_3} \ell(B) = \min_{B \in A_3 \cap A_4} \ell(B) > \ell(B^0)$. Define

$$A_5 = \{\breve{B} \mid \mathrm{dist}(\breve{B}, \mathcal{E}(\Theta^0)) \geq \frac{1}{2} \min_{B \in \mathcal{E}(\Theta^0)} \mathrm{dist}(B, A_3)\}$$

It is easy to see that $A_3 \subseteq A_5$. Then,

$$\min_{B \in A_3 \cap A_4} \ell(B) \geq \min_{B \in A_4 \cap A_5} \ell(B).$$

Note that $A_4 \cap A_5$ is closed, bounded, and nonempty ($0 \in A_4 \cap A_5$), and $\ell(B)$ is a continuous function of $B$. Consequently, there exists at least one minimizer of $\ell(B)$ in $A_4 \cap A_5$. Combining this with the fact that $B \notin A_4 \cap A_5$ for all $B \in \mathcal{E}(\Theta^0)$, we conclude that:

$$\min_{B \in A_3} \ell(B) \geq \min_{B \in A_4 \cap A_5} \ell(B) > \ell(B^0). \qquad \square$$

### B.14 Proof of Corollary 1

*Proof.* By Theorem 1, we know there exists $\lambda_0 > 0$ and $\delta_0 > 0$. For MCP, it can be transformed into quasi-MCP, by reparameterization from Section C.5. Then, combining these results together.

$$0 < \lambda = \lambda_{\mathrm{mcp}} < \lambda_0, 0 < \delta = a\lambda_{\mathrm{mcp}} < \delta_0 \Rightarrow 0 < \lambda_{\mathrm{mcp}} < \lambda_0, 0 < a < \frac{\delta_0}{\lambda_{\mathrm{mcp}}}$$

We could simple set $a_0 : \frac{\delta_0}{\lambda_0}$ and $(\lambda_{\mathrm{mcp}})_0 = \lambda_0$. For SCAD, we just requires the following is satisfied to satisfies the pattern in the proof of Theorem 1.

$$0 < a\lambda_{\mathrm{scad}} < \delta_0, 0 < \frac{\lambda_{\mathrm{scad}}^2(a+1)}{2} < \lambda_0$$

One simple choice is to let

$$a_0 = (\lambda_{\mathrm{scad}})_0 < \min\{\sqrt{\delta_0}, \sqrt{\lambda_0}, 1\}$$

This completes the proof of Corollary 1. $\qquad \square$

# C   Additional Examples and Details

In this appendix, we provide the additional details of derivations, examples, concepts, and discussions referenced in the main paper. These include:

- The derivation of the log-likelihood function for the model in Equation (6) (Appendix C.1).
- A brief introduction to the characterization of the equivalence class $\mathcal{E}(\Theta)$ (Appendix C.2).
- Examples demonstrating that the optimal solution for the least squares loss differs from the optimal solution of the log-likelihood (Appendix C.3).
- An example illustrating the estimation bias when the $\ell_1$ penalty is applied (Appendix C.4).
- The formulations for quasi-MCP, MCP, and SCAD (Appendix C.5).
- The standardization of the random variable $X$ and the dataset $\mathbf{X}$ (Appendix C.6).
- A detailed discussion of Assumptions A and B (Appendix C.7).

## C.1   Log-likelihood of Model (6)

In this subsection, we detail the negative log-likelihood of the model in Equation (6).

$$
\begin{aligned}
\ell_n(B, \Omega) =& -\frac{1}{n} \log \prod_{i=1}^{n} f(\mathbf{x}_i; B, \Omega) \\
=& -\frac{1}{n} \log \prod_{i=1}^{n} \frac{1}{(2\pi)^{p/2} (\det \Sigma(B,\Omega))^{1/2}} \exp\left( -\frac{\mathbf{x}_i^\top \Theta(B,\Omega)\mathbf{x}_i}{2} \right) \\
=& \frac{p}{2} \log 2\pi + \frac{1}{2} \log \det \Sigma(B,\Omega) + \frac{1}{2n} \sum_{i=1}^{n} \mathbf{x}_i^\top \Theta(B,\Omega)\mathbf{x}_i \\
=& \frac{1}{2} \log \det(I-B)^{-\top} \Omega (I-B)^{-1} + \frac{1}{2} \mathrm{Tr}(\Theta(B,\Omega)(\frac{\sum_{i=1}^{n} \mathbf{x}_i^\top \mathbf{x}_i}{n})) + \mathrm{const.} \\
=& \frac{1}{2} \log \det \Omega - \log \det(I-B) + \frac{1}{2} \mathrm{Tr}(\widehat{\Sigma}\Theta(B,\Omega)) + \mathrm{const.} \\
=& \frac{1}{2} \sum_{j=1}^{p} \log w_j^2 + \frac{1}{2} \mathrm{Tr}(\Omega^{-1}(I-B)^\top \widehat{\Sigma}(I-B)) - \log \det(I-B) + \mathrm{const.} \\
=& \frac{1}{2} \sum_{j=1}^{p} \log w_j^2 + \frac{1}{2} \sum_{j=1}^{p} \frac{((I-B)^\top \widehat{\Sigma}(I-B))_{jj}}{\omega_j^2} - \log \det(I-B) + \mathrm{const.}
\end{aligned}
$$

It is easy to know that for any fix $B$, the optimal solution of $(\omega_j^*)^2$ can be written as:

$$
(\omega_j^*)^2 = [(I-B)^\top \widehat{\Sigma}(I-B)]_{jj}
$$

Therefore, optimal solution $\Omega_f(B)$ for any fixed $B$ can be written as:

$$
\Omega_f(B) = \mathrm{diag}((I-B)^\top \widehat{\Sigma}(I-B)) \tag{15}
$$

Let us define profile sample log-likelihood $\ell_n(B)$ as function of $B$ with such optimal $\Omega(B)$ plugged in

$$
\begin{aligned}
\ell_n(B) =& \frac{1}{2} \log \det \mathrm{diag}((I-B)^\top \widehat{\Sigma}(I-B)) - \log \det(I-B) + \mathrm{const.} \\
=& \frac{1}{2} \log \frac{1}{n} \|\mathbf{X}_j - \mathbf{X}B_j\|_2^2 - \log \det(I-B) + \mathrm{const.}
\end{aligned} \tag{16}
$$

Where $\mathbf{X} = (\mathbf{X}_1, \ldots, \mathbf{X}_p) = (\mathbf{x}_1, \ldots, \mathbf{x}_n)^\top$ and corresponding profile population log-likelihood

$$
\begin{aligned}
\ell(B) =& \frac{1}{2} \log \det \mathrm{diag}((I-B)^\top \Sigma(I-B)) - \log \det(I-B) + \mathrm{const.} \\
=& \frac{1}{2} \log \mathbb{E}\|X_j - B_j^\top X\| - \log \det(I-B) + \mathrm{const.}
\end{aligned} \tag{17}
$$

## C.2 Equivalence class $\mathcal{E}(\Theta)$

We provide a brief introduction to the equivalence class $\mathcal{E}(\Theta)$, which has been extensively studied in [2]. We adopt the notation from [2], and further details can be found in that work.

**Definition 6** (topological sort). *a topological sort of a directed graph is an ordering on the nodes, often denoted by $\prec$, such that the existence of a directed edge $X_k \to X_j$ implies that $X_k \prec X_j$ in the ordering.*

Let $\mathcal{P}$ denote the collection of all permutations of the indices $\{1, \ldots, p\}$. For an arbitrary matrix $A$ and any $\pi \in \mathcal{P}$, let $P_\pi A$ represent the matrix obtained by permuting the rows and columns of $A$ according to $\pi$, such that $(P_\pi A)_{ij} = a_{\pi(i)\pi(j)}$.

A DAG $B$ is said to be compatible with permutation $\pi$ if $P_\pi B$ is a lower-triangular matrix, which is equivalent to saying that $X_k \to X_j$ in $B$ implies that $\pi^{-1}(k) > \pi^{-1}(j)$. Similarly, $\pi$ is also called compatible with $B$.

For any positive definite matrix $\Theta$ and $\pi \in \mathcal{P}$, the matrix $P_\pi \Theta$ represents the same covariance structure as $\Theta$, up to a reordering of the variables. The Cholesky decomposition of $P_\pi(\Theta)$ can be uniquely written as:

$$P_\pi \Theta = (I - L)D^{-1}(I - L)^\top = \Theta_f(L, D),$$

where $L$ is strictly lower triangular and $D$ is diagonal. By Lemma 8 in [2], the following holds:

$$P_\pi \Theta(L, D) = \Theta(P_\pi L, P_\pi D) \qquad \forall \pi \in \mathcal{P}.$$

Therefore,

$$\Theta = \Theta_f(P_{\pi^{-1}}L, P_{\pi^{-1}}D).$$

For each $\pi$, we define:

$$\widetilde{B}(\pi) := P_{\pi^{-1}}L,$$
$$\widetilde{\Omega}(\pi) := P_{\pi^{-1}}D.$$

This suggests that for any $\pi \in \mathcal{P}$, there exists a pair $(\widetilde{B}(\pi), \widetilde{\Omega}(\pi)) \in \mathcal{E}(\Theta)$, where $\widetilde{B}(\pi)$ can be uniquely determined based on the permutation $\pi$ and $\Theta$ [2]. It is important to emphasize that different permutations, $\pi_1 \neq \pi_2$, can still result in the same pairs, i.e., $(\widetilde{B}(\pi_1), \widetilde{\Omega}(\pi_1)) = (\widetilde{B}(\pi_2), \widetilde{\Omega}(\pi_2))$. Furthermore, this indicates that for any $(B, \Omega) \in \mathcal{E}(\Theta)$, there exists at least one permutation $\pi$ such that $(B, \Omega) = (\widetilde{B}(\pi), \widetilde{\Omega}(\pi))$. Moreover, it turns out that the collection of pair of $(\widetilde{B}(\pi), \widetilde{\Omega}(\pi))$ forms the entire equivalence class $\mathcal{E}(\Theta)$.

**Lemma 9** (Lemma 1, [2]). *Suppose $\Sigma$ is a positive definite covariance matrix and $\Theta = \Sigma^{-1}$. Then,*

$$\mathcal{E}(\Theta) = \{(P_{\pi^{-1}}L, P_{\pi^{-1}}D) : P_\pi \Theta = \Theta_f(L, D), \pi \in \mathcal{P}\}$$
$$= \{(\widetilde{B}(\pi), \widetilde{\Omega}(\pi)) : \pi \in \mathcal{P}\}$$

This result indicates that the size of $\mathcal{E}(\Theta)$ is at most $p!$, which is large but finite.

## C.3 LS loss vs. log-likelihood

The following examples show that when the variances are unequal, the LS loss will not in general have the same minimizers as the log-likelihood. The first example is just Example 1 in [33]. Suppose $(X_1, X_2)$ is distributed according to the following linear SEM with unequal variances:

$$X_1 = \epsilon_1, \qquad X_2 = -\frac{X_1}{2} + \epsilon_2, \qquad \epsilon_1 \sim N(0, 1), \qquad \epsilon_2 \sim N(0, 1/4).$$

Thus

$$B^0 = \begin{pmatrix} 0 & -1/2 \\ 0 & 0 \end{pmatrix}, \qquad \Omega^0 = \begin{pmatrix} 1 & 0 \\ 0 & 1/4 \end{pmatrix}, \qquad \Sigma^0 = \Sigma_f(B^0, \Omega^0) = \begin{pmatrix} 1 & -1/2 \\ -1/2 & 1/2 \end{pmatrix}$$

and also

$$\mathcal{E}(\Theta^0) = \mathcal{E}_{\min}(\Theta^0) = \{B^0, B_1\}$$

where

$$B_1 = \begin{pmatrix} 0 & 0 \\ -1 & 0 \end{pmatrix}, \quad \Omega_1 = \begin{pmatrix} 1 & 0 \\ 0 & 1 \end{pmatrix}.$$

Moreover, $\ell(B^0, \Omega^0) = \ell(B_1, \Omega_1)$, since both SEM represent the same covariance.

But it turns out that $\mathbb{E}[\|X - B_1^\top X\|^2] < \mathbb{E}[\|X - (B^0)^\top X\|^2]$: More precisely, it is easy to check that

$$\mathbb{E}[\|X - B_1^\top X\|^2] = \mathrm{Tr}((I - B_1)^\top \Sigma^0 (I - B_1)) = 1,$$
$$\mathbb{E}[\|X - (B^0)^\top X\|^2] = \mathrm{Tr}((I - B^0)^\top \Sigma^0 (I - B^0)) = 5/4,$$

and moreover $B_1$ is the global minimizer of the LS loss $\mathbb{E}[\|X - B^\top X\|^2]$. It follows that when the variances are different, the log-likelihood and LS loss have different global minimizers.

Similar calculations can be carried out for $d > 2$, but are tedious owing to the size of $\mathcal{E}(\Theta)$. For example, here is an example of an SEM over 3 nodes such that the LS loss has a different set of global minimizers, but also the LS-global minimizer has *more* edges than the sparsest Markov representation:

$$B^0 = \begin{pmatrix} 0 & 0 & -3/10 \\ 0 & 0 & -2 \\ 0 & 0 & 0 \end{pmatrix}, \quad \Omega^0 = \begin{pmatrix} 7 & 0 & 0 \\ 0 & 3 & 0 \\ 0 & 0 & 2 \end{pmatrix}, \quad \Sigma^0 = \Sigma_f(B^0, \Omega^0) = \begin{pmatrix} 7 & 0 & -2 \\ 0 & 3 & -5 \\ -2 & -5 & 10 \end{pmatrix}.$$

For this model, LS loss selects the following SEM with 3 edges:

$$B_1 = \begin{pmatrix} 0 & 0 & 0 \\ -1.197 & 0 & -1.589 \\ -0.7532 & 0 & 0 \end{pmatrix}.$$

We have $B^0 \in \mathcal{E}_{\min}(\Theta^0)$, but $B_1 \notin \mathcal{E}_{\min}(\Theta^0)$.

### C.4 Estimation bias under $\ell_1$

We provide an example showing that when the $\ell_1$ penalty is applied, the estimation becomes biased. Therefore, $\ell_1$ should not be used. Consider the following linear Structural Equation Model (SEM):

$$\begin{cases} X_1 = \mathcal{N}(0, 1) \\ X_2 = X_1 + \mathcal{N}(0, \sigma^2) \end{cases}$$

If the topological sort is known, i.e., $X_1 \to X_2$, and an $\ell_1$ penalty is used for minimizing the negative log-likelihood:

$$\min_a \log \mathbb{E}[\|X_2 - aX_1\|_2^2] + \log \mathbb{E}[\|X_1\|_2^2] + \lambda|a|$$

Ideally, we would expect $a = 1$ to be the minimal solution to the loss function. However, the problem is equivalent to:

$$\log((1 - a)^2 + \sigma^2) + \lambda|a|$$

It is clear that $a = 1$ is not the minimal solution to the loss function, as the derivative at $a = 1$ is nonzero for any $\lambda > 0$, indicating that the $\ell_1$ penalty leads to a biased estimator. This bias does not occur when using MCP or SCAD with appropriate hyperparameters.

### C.5 quasi-MCP, MCP and SCAD

We present the formulas for quasi-MCP, MCP, and SCAD, and demonstrate that quasi-MCP and MCP are equivalent.

quasi-MCP $\quad p_{\lambda,\delta}(t) = \lambda\left[\left(|t| - \frac{t^2}{2\delta}\right)\mathbb{1}(|t| < \delta) + \frac{\delta}{2}\mathbb{1}(|t| > \delta)\right]$

MCP $\quad p_{\lambda,a}^{MCP}(t) = \mathbb{1}(|t| < a\lambda)\left(\lambda|t| - \frac{t^2}{2a}\right) + \mathbb{1}(|t| \geq \lambda a)\frac{\lambda^2 a}{2}$

SCAD $\quad p_{\lambda,a}^{SCAD}(t) = \lambda|t|\mathbb{1}(|t| < \lambda) + \mathbb{1}(\lambda < |t| < a\lambda)\frac{2a\lambda|t| - t^2 - \lambda^2}{2(a - 1)} + \mathbb{1}(|t| \geq \lambda a)\frac{\lambda^2(a + 1)}{2}$

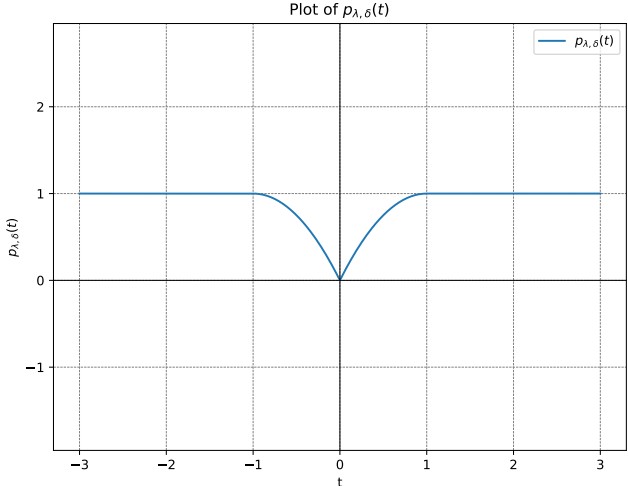

Figure 4: The plot of $p_{\lambda,\delta}(t)$ with $\lambda = 2, \delta = 1$

It is worth noting that if we set $\delta$ as $a\lambda$ in quasi-MCP, then $p_{\lambda,a\lambda}(t) = p_{\lambda,a}^{MCP}(t)$. In another way, if we set $a = \frac{\delta}{\lambda}$ in MCP, then $p_{\lambda,\frac{\delta}{\lambda}}^{MCP}(t) = p_{\lambda,a}(t)$. Thus, quasi-MCP and MCP are equivalent to each other.

### C.6 Standardization of $X$ and $\mathbf{X}$

We present the formulas for the standardization of $X$ and the standardization of the corresponding dataset $\mathbf{X}$.

Let $\sigma_i^2 := \mathrm{Var}(X_i)$ and $D := \mathrm{diag}(\sigma_1, \ldots, \sigma_p)$. Denote the standardized version of $X$ as $X_{\mathrm{std}}$, which can be expressed as:

$$X_{\mathrm{std}} = D^{-1}(X - \mathbb{E}X).$$

For $\mathbf{X} \in \mathbb{R}^{n \times p}$, we can write $\mathbf{X} = (\mathbf{X}_{ij}) = (\mathbf{x}_1, \ldots, \mathbf{x}_n)^\top = (\mathbf{X}_1, \ldots, \mathbf{X}_p)$, and define sample average for node $j$ as $\widehat{\mu}_j$

$$\widehat{\mu}_j = \frac{1}{n} \sum_{i=1}^n \mathbf{X}_{ij} \qquad \forall j \in [p].$$

Next, we define the sample variance for node $j$ as

$$\widehat{\sigma}_j^2 = \frac{1}{n-1} \sum_{i=1}^n (\mathbf{X}_{ij} - \widehat{\mu}_j)^2.$$

The diagonal matrix of sample standard deviations is then

$$\widehat{D} = \mathrm{diag}(\widehat{\sigma}_1, \ldots, \widehat{\sigma}_p).$$

Finally, we standardize $\mathbf{X}$ by subtracting the sample means and scaling by the inverse of $\widehat{D}$:

$$\mathbf{Z} = [\mathbf{X} - \mathbf{1}_n \cdot (\widehat{\mu}_1, \ldots, \widehat{\mu}_p)]\widehat{D}^{-1},$$

where $\mathbf{1}_n \in \mathbb{R}^n$ is an $n$-dimensional vector with all entries equal to $1$.

### C.7 Discussion of Assumption A, B

In this subsection, we provide a more detailed discussion of Assumptions A and B.

First, it is important to emphasize that if $p_{\lambda,\delta}$ is replaced by the $\ell_0$ penalty in (12) or (14), then Assumptions A and B can be omitted, and all the results still hold. In this case, the proof would be significantly simplified. However, the use of the differentiable quasi-MCP, in contrast to the $\ell_0$ penalty, introduces substantial complications, necessitating some additional assumptions that are fundamentally different from the results in [8]. Specifically, Assumptions A and B are exactly what is required to make the problem suitable for gradient-based optimization.

Assumption A is not restrictive and is satisfied by all identifiable models, including linear Gaussian models, generalized linear models with continuous output [66], binary output [74, 4], and most exponential families. However, the requirement for the finiteness of the equivalence class can be relaxed. What is truly needed is that the minimal nonzero edge has sufficient "signal," i.e.,

$$\min_{(\psi,\xi)\in\mathcal{E}(\psi^0,\xi^0)} \min_{\{(i,j):B(\psi)_{ij}\neq 0\}} |B(\psi)|_{ij} > 0$$

This is trivially true when $|\mathcal{E}(\psi^0,\xi^0)|$ is finite. When $|\mathcal{E}(\psi^0,\xi^0)|$ is infinite, each $|B(\psi)|_{ij}$ could be positive, but it is possible $\liminf_{(\psi,\xi)\in\mathcal{E}(\psi^0,\xi^0)} \min_{\{(i,j):B(\psi)_{ij}\neq 0\}} |B(\psi)|_{ij} = 0$, because $|B(\psi)|_{ij}$ can be arbitrarily small. The $\ell_0$ penalty deals with this with its discontinuity at zero, whereas the continuity of quasi-MPC makes this more challenging. This is the cost of differentiability, which we argue is worthwhile

Assumption B is a standard assumption in the optimization literature [7] and is generally quite weak. Moreover, it is almost necessary because quasi-MCP cannot exactly count the number of edges in $B(\psi)$. The magnitude of the quasi-MCP penalty does not directly reveal the number of edges. This is the trade-off for replacing the $\ell_0$ penalty with a fully differentiable sparsity-inducing penalty. Finally, it is worth noting that this assumption can also be relaxed: what is truly required is that for any $\epsilon > 0$, there exists $\delta > 0$ such that

$$\ell(\psi,\xi) - \ell(\psi^0,\xi^0) > \delta \quad \text{for all } \{(\psi,\xi) \mid \text{dist}((\psi,\xi),\mathcal{E}(\psi^0,\xi^0)) > \epsilon\}.$$

In other words, we require a loss gap when $(\psi,\xi)$ is not in $\mathcal{E}(\psi^0,\xi^0)$. This can be inferred from Assumption B.

The following is the proof when $p_{\lambda,\delta}$ is replaced with the $\ell_0$ penalty.

**Proof when $p_{\lambda,\delta}$ is replaced with $\ell_0$:**

*Proof.* We can also assume that $|\mathcal{E}_{\min}(\psi^0,\xi^0)| = 1$, meaning $\mathcal{E}_{\min}(\psi^0,\xi^0) = \{(\psi^0,\xi^0)\}$. In other words, there is a unique element in the minimal equivalence class. This is because any element in $\mathcal{E}_{\min}(\psi^0,\xi^0)$ is indistinguishable based on the score function, i.e., the value of $\ell(\psi,\xi)$ and the penalty for the number of edges in $B(\psi)$. Our objective is to find this unique element by solving equation (12) in the paper, which simplifies the proof.

When $s_{B(\psi^0)} = 0$, the result is straightforward:

$$\ell(\psi^0,\xi^0) + s_{B(\psi^0)} \leq \ell(\psi,\xi) + s_{B(\psi)}.$$

Now, let us consider the more general case where $s_{B(\psi^0)} > 0$ and divide the parameter space into three regions:

$$A_1 = \{(\psi,\xi) \mid s_{B(\psi)} > s_{B(\psi^0)}\}, \quad A_2 = \{(\psi,\xi) \mid s_{B(\psi)} = s_{B(\psi^0)}\}, \quad A_3 = \{(\psi,\xi) \mid s_{B(\psi)} < s_{B(\psi^0)}\}.$$

**Case 1: Consider** $A_1$. Since $\ell(\psi^0,\xi^0) \leq \ell(\psi,\xi)$, the following holds for any $\lambda > 0$:

$$\ell(\psi^0,\xi^0) + \lambda s_{B(\psi^0)} < \ell(\psi,\xi) + \lambda s_{B(\psi)} \quad \forall(\psi,\xi) \in A_1.$$

**Case 2: Consider** $A_2$. Since $|\mathcal{E}_{\min}(\psi^0,\xi^0)| = 1$, it follows that for all $(\psi,\xi) \in \mathcal{E}(\psi^0,\xi^0)$ and $(\psi,\xi) \neq (\psi^0,\xi^0)$, we have $s_{B(\psi)} > s_{B(\psi^0)}$. Therefore, for all $(\psi,\xi) \in A_2$, it holds that $\ell(\psi^0,\xi^0) < \ell(\psi,\xi)$. Consequently, for any $\lambda > 0$:

$$\ell(\psi^0,\xi^0) + \lambda s_{B(\psi^0)} < \ell(\psi,\xi) + \lambda s_{B(\psi)} \quad \forall(\psi,\xi) \in A_2.$$

**Case 3: Consider** $A_3$. We need to prove that:

$$\ell(\psi^0,\xi^0) + \lambda s_{B(\psi^0)} < \ell(\psi,\xi) + \lambda s_{B(\psi)} \quad \forall(\psi,\xi) \in A_3.$$

This is equivalent to showing that there exists a positive $\lambda$ such that:

$$\lambda < \frac{\ell(\psi, \xi) - \ell(\psi^0, \xi^0)}{s_{B(\psi^0)} - s_{B(\psi)}} \quad \forall (\psi, \xi) \in A_3.$$

Since $s_{B(\psi^0)}$ is the minimal number of edges in the equivalence class, any $(\psi, \xi) \in A_3$ corresponds to $B(\psi)$ with a number of edges strictly less than $s_{B(\psi^0)}$. This implies that:

$$\ell(\psi, \xi) - \ell(\psi^0, \xi^0) > 0 \quad \forall (\psi, \xi) \in A_3.$$

Furthermore, we have $1 \leq s_{B(\psi^0)} - s_{B(\psi)} \leq s_{B(\psi^0)}$, which implies that there exists a small but positive $\lambda$ that satisfies the inequality.

$\square$

# D    Experiment Details

In this section, we provide all the details about the experiments. These include: (1) the types of graphs used, (2) the process for generating the samples, (3) the baseline methods we compare against and where to find the code for these methods, (4) the implementation details of our method and how to replicate the results, and (5) the metrics used to evaluate the estimation.

## D.1    Experimental Setting

In this section, we outline the process for generating graphs and data for Structural Equation Models (SEMs) in (2). For each model, a random graph $G$ is generated using one of two types of random graph models: Erds-Rényi (ER) or Scale-Free (SF). The models are specified to have, on average, $kp$ edges, where $k \in \{1, 2, 4\}$. These configurations are denoted as ER$k$ or SF$k$, respectively.

- *Erds-Rényi* (ER), Random graphs whose edges are add independently with equal probability. We simulated models with $p, 2p$ and $4p$ edges (in expectation) each, denoted by $ER1, ER2$, and $ER4$ respectively.
- Scale-free network(SF). Network simulated according to the preferential attachment process [3]. We simulated scale-free network with $p, 2p$ and $4p$ edges and $\beta = 1$, where $\beta$ is the exponent used in the preferential attachment process.

**Linear SEMs.**    Given a random DAG $B \in \{0, 1\}^{p \times p}$ from one of these two graph models, edge weights were assigned independently from Unif($[-1.5, -0.5] \cup [0.5, 1.5]$) to obtain a weight matrix $B \in \mathbb{R}^{p \times p}$. Given $B$, we sampled $X = B^\top X + z \in \mathbb{R}^p$ according to:

- *Gaussian noise* with unequal variance (Gauss-NV): $z_i \sim \mathcal{N}(0, \sigma_i^2), i = 1, \ldots, p$ where $\sigma_i \sim \text{Unif}[0.1, 0.7]$

We chose to set $\sigma_i$, the noise variances in our models, to be relatively smaller compared to the settings used in previous studies such as [73], [41], and [4]. This decision aims to mitigate the potential exploitation of accumulated variance along the topological sort, as highlighted in [55].

**Generalized Linear Model with Binary Output**    Given a random DAG $B \in \{0, 1\}^{p \times p}$ from one of these two graph models, edge weights were assigned independently from Unif($[-1.5, -0.5] \cup [0.5, 1.5]$) to obtain a weight matrix $B \in \mathbb{R}^{p \times p}$. Given $B$, we sample $X_j$ according to the following

$$X_j = \text{Bernoulli}(\exp(B_j^\top X)/(1 + \exp(B_j^\top X))) \quad j = 1, \ldots, p$$

where $B_j$ is $j$-th column of $B$. The corresponding negative log-likelihood function:

$$s(B; \mathbf{X}) = \frac{1}{n} \sum_{i=1}^{p} \mathbf{1}_n^\top \left( \log(\mathbf{1}_n + \exp(\mathbf{X}B)) - \mathbf{X}_i \circ (\mathbf{X}B) \right)$$

where $\mathbf{X} = (\mathbf{X}_{ij}) = (\mathbf{x}_1, \ldots, \mathbf{x}_n)^\top = (\mathbf{X}_1, \ldots, \mathbf{X}_p)$

**Nonlinear Models with Neural Networks.** We primarily follow the nonlinear setting described in Zheng et al. [74]. Given $G$, we simulate the SEM as follows:

$$X_j = f_j(X_{\text{pa}(j)}) + N_j \qquad \forall j \in [p],$$

where $N_j \sim \mathcal{N}(0, \sigma_i^2)$ and $\sigma_i \sim \text{Uni}[0.1, 1]$. Here, $f_j$ is a randomly initialized MLP with one hidden layer of size 100 and sigmoid activation. It is worth noting that the score function used in nonlinear-NOTEARS [74] is least square loss:

$$s(f, \mathbf{X}) = \frac{1}{2n} \sum_{i=1}^{p} \|\mathbf{x}_i - \widehat{f}_i(\mathbf{X})\|^2,$$

where each $\widehat{f}_i$ is an MLP with one hidden layer of size 30 and sigmoid activation.

**Simulation** We generated random datasets $\mathbf{X} \in \mathbb{R}^{n \times p}$ by sampling rows i.i.d. from the models described above. For each simulation, we produced datasets with $n$ samples across graphs with $p$ nodes.

- **Linear Model:** $p = \{10, 20, 50, 70, 100\}$, $k = \{1, 2, 4\}$, $n = 1000$ and graph types = {ER, SF}.

- **Generalized Linear Model:** $p = \{10, 20, 40\}$, $k = \{1, 2\}$, $n = 10000$ and graph types = {ER, SF}.

- **Nonlinear Model:** $p = \{10, 20, 40\}$, $k = \{1, 2\}$, $n = 1000$ and graph types = {ER, SF}.

For each dataset, we applied several structural learning algorithms, including fast greedy equivalence search (FGES [52]), constraint-based methods (PC [60]), NOTEARS [73, 74] (using least squares loss), GOLEM [41] (using NLL with $\ell_1$ penalty), VarSort [55], causal additive models (CAM [9]), LOGLL(-NOTEARS/DAGMA)-SAMPLE (utilizing the sample covariance matrix $\widehat{\Sigma}$), LOGLL(-NOTEARS/DAGMA)-POPULATION (using the population covariance matrix $\Sigma$) and exact method (EXACT-SEARCH). Implementation details are provided in the following paragraph. After running the algorithms, a post-processing threshold of $0.3$ was applied to the estimated matrix $B_{\text{est}}$ to prune small values, following the methodology in [73, 74].

**Implementation** The implementation details of baseline are listed below:

- Fast Greedy Equivalence Search (FGES [52]) is based on greedy search and assumes linear dependency between variables. The implementation is based on the `py-tetrad` package, available at `https://github.com/cmu-phil/py-tetrad`. We use `BIC` as the score function with default parameters.

- PC [60] is constraint-based method and based on uses conditional independence induced by causal relationships to learn those causal relationships. The implementation is based on the `py-tetrad` package, available at `https://github.com/cmu-phil/py-tetrad`. We use Fisher-$Z$ test with $\alpha = 0.5$.

- NOTEARS[73, 74] is the continuous DAG learning algorithm using least square loss with $\ell_1$ regularization. It is implemented in python: `https://github.com/xunzheng/notears`.

- GOLEM [41] is implemented using Python and TensorFlow. The code is available `https://github.com/ignavierng/golem`.

- VarSort [55] is based on the observation that variances tend to accumulate along the topological sort. It uses Lasso [61] to recover the coefficients. The code is implemented in Python and is available `https://github.com/Scriddie/Varsortability`.

- DAGMA[4] is a continues DAG learning algorithm with better accuracy and faster computational speed. It also use least square loss with $\ell_1$ penalty as NOTEARS. The implementation is available at `https://github.com/kevinsbello/dagma`.

- Causal additive model (CAM [9]) learns an addititve SEM by leveraging efficient nonparametric regression techiques and greedy search over edges. The code is implemented in R, and avaiable at `https://rdrr.io/cran/CAM/man/CAM.html`

- LOGLL(-NOTEARS/DAGMA)-SAMPLE/POPULATION is our approach, which modifies the original NOTEARS or DAGMA algorithm by replacing its scoring function. Instead of using the least squares loss with an $\ell_1$ penalty, it employs a log-likelihood function that includes a quasi-MCP penalty, as defined in (10). For LOGLL(-NOTEARS/DAGMA)-SAMPLE, we use the sample covariance matrix $\hat{\Sigma}$ in the score function. In contrast, LOGLL(-NOTEARS/DAGMA)-POPULATION uses the true covariance matrix $\Sigma$ as a baseline approach. In this paper, LOGLL-NOTEARS refers to solving (12) using NOTEARS with NLL and quasi-MCP as the score function, while LOGLL-DAGMA refers to solving (12) using DAGMA with NLL and quasi-MCP as the score function.

- EXACT-SEARCH is used to indicate that the optimization problem (12) is solved exactly. This approach is feasible only for small graphs, where we attempt to calculate all possible configurations $\tilde{B}(\pi)$ as defined in Section C.2. These calculations can be performed using Cholesky Decomposition or Ordinary Least Squares (OLS). The label POPULATION signifies that the operation is based on the population covariance matrix $\Sigma$, while SAMPLE denotes that it is based on the sample covariance matrix $\widehat{\Sigma}$. The Structural Hamming Distance (SHD) for EXACT-SEARCH is calculated on an average basis. This involves identifying the set $\mathcal{M}_{\min}(\Theta)$ or $\mathcal{M}_{\min}(\widehat{\Theta})$, calculating the SHD for each DAG within this set, and then computing the average SHD.

### D.2 Implementation of LogLL(-NOTEARS/DAGMA)-population/sample

**Linear model**   There are two main challenges in solving (14). The first challenge is that (12) is a highly nonconvex optimization problem and is sensitive to initialization. If we randomly initialize or set the initialization to zero, as done in [73, 74, 41], LOGLL-NOTEARS/DAGMA often gets stuck at a local optimal solution. The second challenge arises from Theorem 1, where we are advised to select $\lambda$ and $\delta$ such that $0 < \lambda < \lambda_0$ and $0 < \delta < \delta_0$. Theoretically, smaller values for $\lambda$ and $\delta$ should be used to adhere to the theorem's guidelines, however, in practice, solving the optimization problem (12) to global optimality is not always feasible.

To address the first challenge, we adopt the approach from Ng et al. [41]. We first run NOTEARS (with least squares loss and $\ell_1$ penalty) or DAGMA (with least squares loss and $\ell_1$ penalty) to obtain a "good" initialization point. Then, we apply LOGLL(-NOTEARS/DAGMA)-POPULATION/SAMPLE to obtain the final output.

To address second challenge, we use warm starts. We begin with larger values for $\lambda$ and $\delta$ and solve (12) using LOGLL-NOTEARS/DAGMA to obtain an initial $B_{\text{est}}$. We then reduce $\lambda$ and $\delta$ by a factor of $\gamma < 1$ and use the previous output as the starting point for the next iteration of LOGLL-NOTEARS/DAGMA. This process is repeated until the negative log-likelihood $\ell_n(B_{\text{est}}, \Omega_{\text{est}})$ begins to increase. This iterative approach helps to refine the solutions gradually, ensuring that each step starts from a potentially better approximation, as formally outlined in Algorithm 1.

In Algorithm 1, we detail the complete implementation of LOGLL(-NOTEARS/DAGMA)-SAMPLE. By replacing $\widehat{\Sigma}$ with $\Sigma$ and substituting $\ell_n$ with $\ell$, this algorithm is adapted to the full implementation of LOGLL(-NOTEARS/DAGMA)-POPULATION.

It turns out that LOGLL-DAGMA outperforms LOGLL-NOTEARS in our experiments. Therefore, we present the results from LOGLL-DAGMA.

**Nonlinear model**   We utilize LOGLL-NOTEARS, which uses the same optimization framework in [74] with replacement of least square loss to negative log-likelihood loss, we keep other hyperparameter unchanged. The score function (negative log-likelihood) we use

$$s_{NLL}(f, \mathbf{X}) = \frac{1}{2n} \sum_{i=1}^{d} \log\left(\|\mathbf{x}_i - \widehat{f_i}(\mathbf{X})\|^2\right)$$

Here $\hat{f}_i$ is $i$-th MLP with one hidden layer of size 40 and sigmoid activation.

**Standardized data Z**   Although it has been shown that the log-likelihood score is scale-invariant for the linear model with Gaussian noise (see Theorem 3), it was observed that using standardized

---

**Algorithm 1:** Full Implementation

---

**Input:** Sample covariance $\widehat{\Sigma}$, decay factor $\gamma \in (0, 1)$, and $\lambda, \delta$, initial point $(B_{\text{in}}, \Omega_{\text{in}})$, initial loss $\ell_{\text{in}}$ (typically very large)

`// initial point` $(B_{\text{in}}, \Omega_{\text{in}})$ `is obtained from NOTEARS or DAGMA`

**Output:** $(B_{\text{est}}, \Omega_{\text{est}})$

**1 while** *True* **do**
**2**     Solve LOGLL-NOTEARS or LOGLL-DAGMA with input $(B_{\text{in}}, \Omega_{\text{in}})$, and get output $(B_{\text{out}}, \Omega_{\text{out}})$
**3**     Calculate $\ell_n(B_{\text{out}}, \Omega_{\text{out}})$
**4**     **if** $\ell_{in} > \ell_n(B_{out}, \Omega_{out})$ **then**
**5**        $\ell_{\text{in}} \leftarrow \ell_n(B_{\text{out}}, \Omega_{\text{out}})$
**6**        $\lambda \leftarrow \gamma\lambda$
**7**        $\delta \leftarrow \gamma\delta$
**8**        $(B_{\text{in}}, \Omega_{\text{in}}) \leftarrow (B_{\text{out}}, \Omega_{\text{out}})$
**9**     **else**
**10**        **return** $(B_{\text{in}}, \Omega_{\text{in}})$
**11**     **end**
**12 end**

---

data $\mathbf{Z}$ makes solving the optimization problem (12) significantly more challenging. For the LOGLL-NOTEARS implementation, the LBFGS-B algorithm fails to produce meaningful solutions. As a result, we replaced LBFGS-B with ADAM [27], an optimizer better suited for handling the difficulties of standardized data, to solve the subproblem in NOTEARS. Alternatively, directly using LOGLL-DAGMA is another effective way to address this challenge. Empirically, we find that setting $\gamma = 0.8$, $\lambda = 0.4$, and $\delta = 0.2$ usually serves as a good choice for the parameters in our optimization procedures.

### D.3 Metrics

We evaluate the performance of each algorithm with the following three metrics:

- **Structure Hamming distance (SHD)**: A standard benchmark in the structure learning literature that counts the total number of edges additions, deletions, and reversals needed to convert the estimated graph into the true graph. Since our model specified in (6) is unidentifiable, the Structural Hamming Distance (SHD) is calculated with respect to the completed partially directed acyclic graph (CPDAG) of the ground truth and $B_{\text{est}}$. We utilize the code from Zheng et al. [75].

- **Times:** The amount of time the algorithm takes to run, measured in seconds. This metric is used to evaluate the speed of the algorithms.

# E    Additional Results

## E.1    Linear Model (SHD)

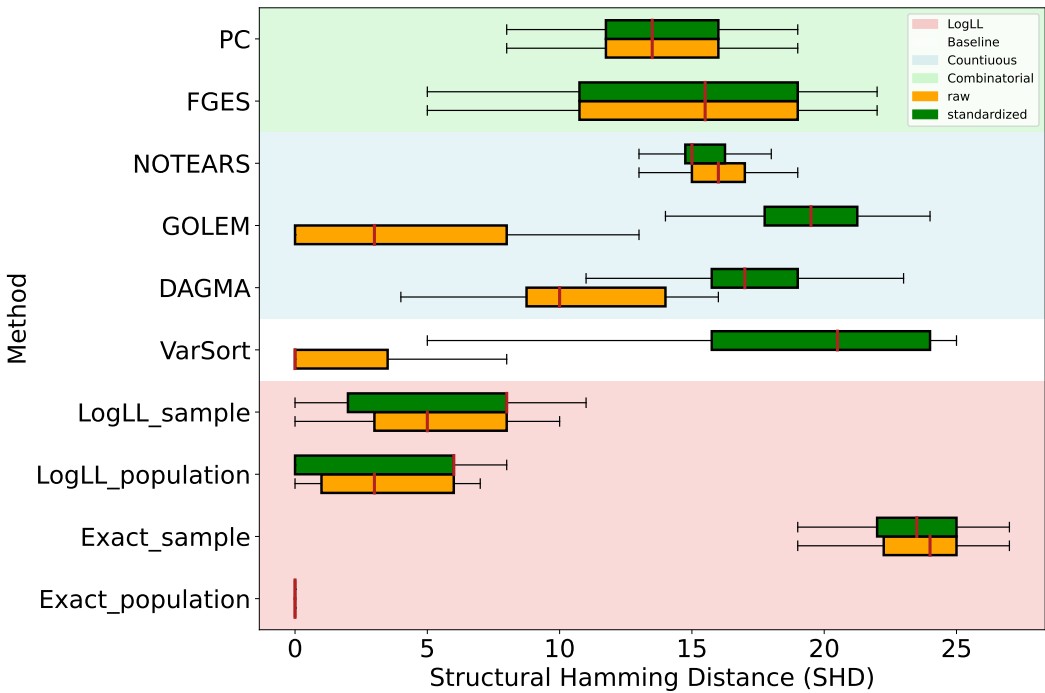

Figure 5: Structural Hamming Distance (SHD, with lower values indicating better performance) between Markov equivalence classes (MEC) of recovered and ground truth graphs for ER-2 graphs with 8 nodes. Here Exact-search is added to illustrate Theorem 3. Standardization does not affect the DAG structure if the optimization (10) can be solved globally. Both Exact-sample and Exact-population produce the same DAG structure for raw data $\mathbf{X}$ and standardized data $\mathbf{Z}$. When the population covariance matrix is known, $\mathcal{E}_{\min}(\Theta^0) = \mathcal{M}(G^0)$, resulting in an SHD of zero. The poor performance of Exact-sample can be attributed to the lack of thresholding applied to the coefficients recovered from Ordinary Least Squares (OLS). Since $\widehat{\Sigma}$ is only an approximation of $\Sigma$, coefficients derived from OLS based on different permutations $\pi$ may shift from zero to nonzero, even though such coefficients might be very small. However, since Exact is impractical for real-world applications, we use this example primarily for illustrative purposes, and thus no threshold is applied to this method.

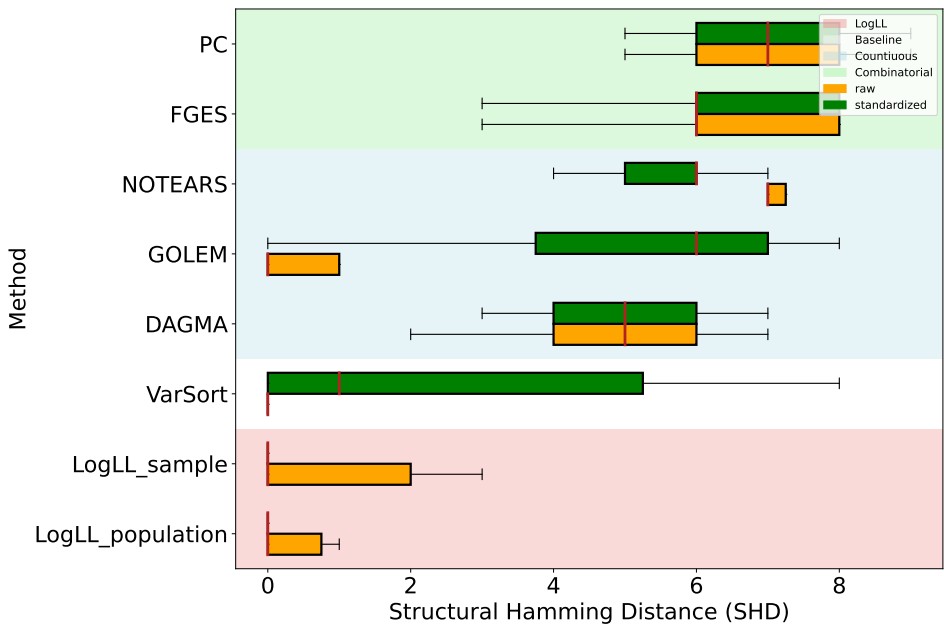

Figure 6: Comparison of raw (orange) vs. standardized (green) data. Structural Hamming Distance (SHD, with lower values indicating better performance) between Markov equivalence classes (MEC) of recovered and ground truth graphs for ER-2 graphs with 5 nodes

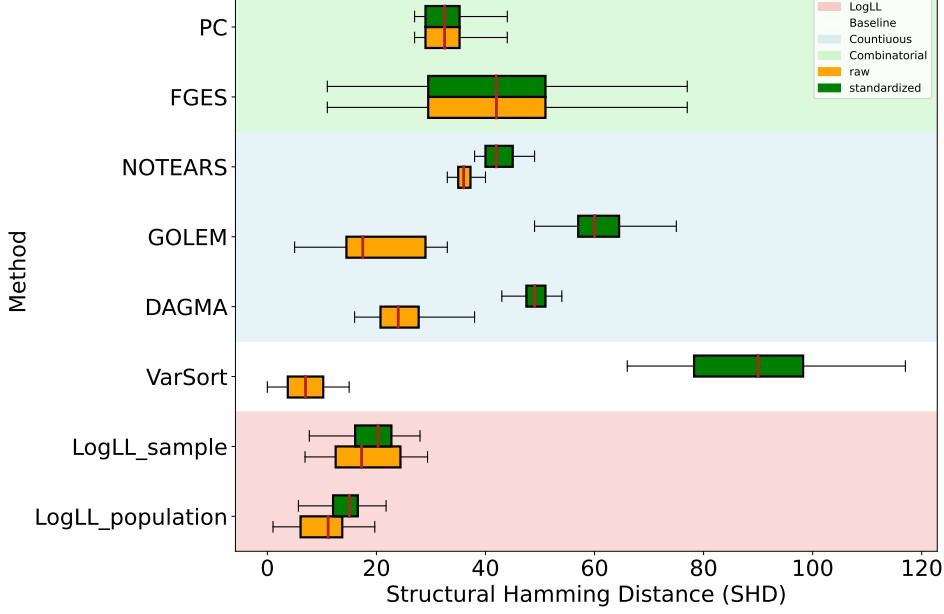

Figure 7: Comparison of raw (orange) vs. standardized (green) data. Structural Hamming Distance (SHD, with lower values indicating better performance) between Markov equivalence classes (MEC) of recovered and ground truth graphs for ER-2 graphs with 20 nodes

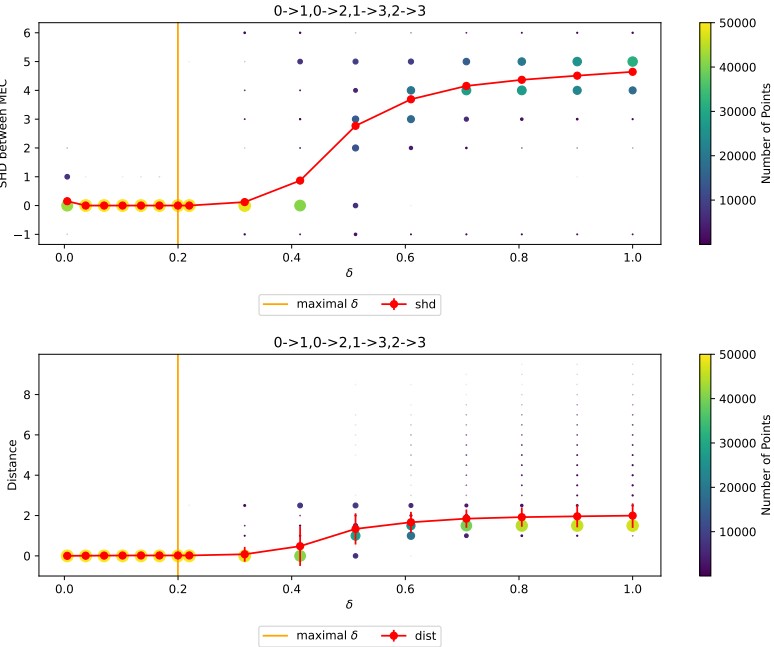

Figure 8: Graph: structure $X_0 \rightarrow X_1, X_0 \rightarrow X_2, X_1 \rightarrow X_3, X_2 \rightarrow X_3$. For $0 < \delta < \delta_0$, the estimated $(B_{\text{est}}, \Omega_{\text{est}}) \in \mathcal{E}_{\min}(\Theta^0)$ because SHD and distance are closed to 0.

.

Figure 9: Graph: structure $X_0 \rightarrow X_1, X_2 \rightarrow X_1$. For $0 < \delta < \delta_0$, the estimated $(B_{\text{est}}, \Omega_{\text{est}}) \in \mathcal{E}_{\min}(\Theta^0)$ because SHD and distance are closed to 0.

.

## E.2 Linear Model (Time)

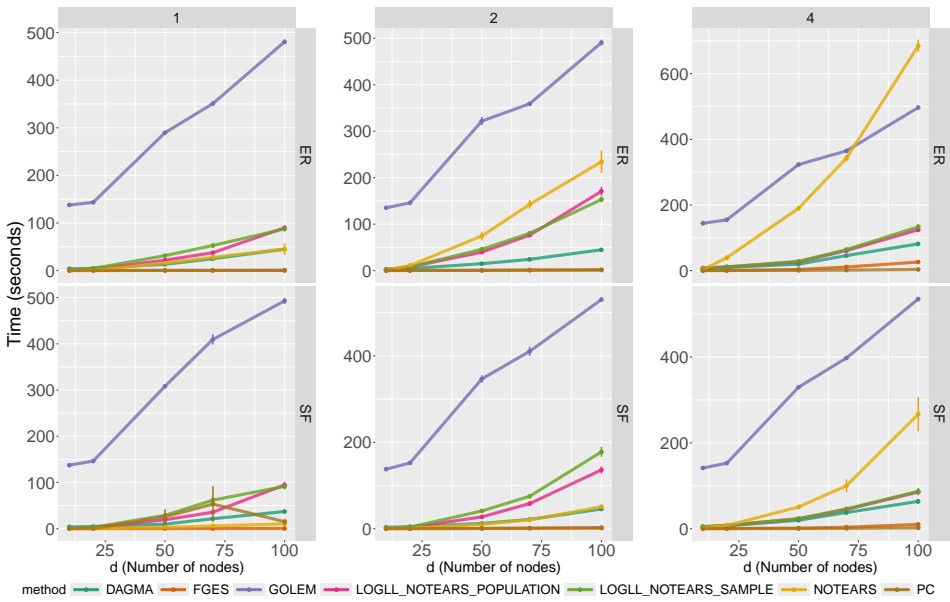

Figure 10: Results in term of Time. Lower is better. Column: $k = \{1, 2, 4\}$. Row: random graph types. {ER,SF}-$k$ = {Scale-Free,Erds-Rényi } graphs with $kd$ expected edges. Here $d = \{10, 20, 50, 70, 100\}$, $n = 1000$. Standard error is removed for better visualization. It is for different methods on raw data $\mathbf{X}$

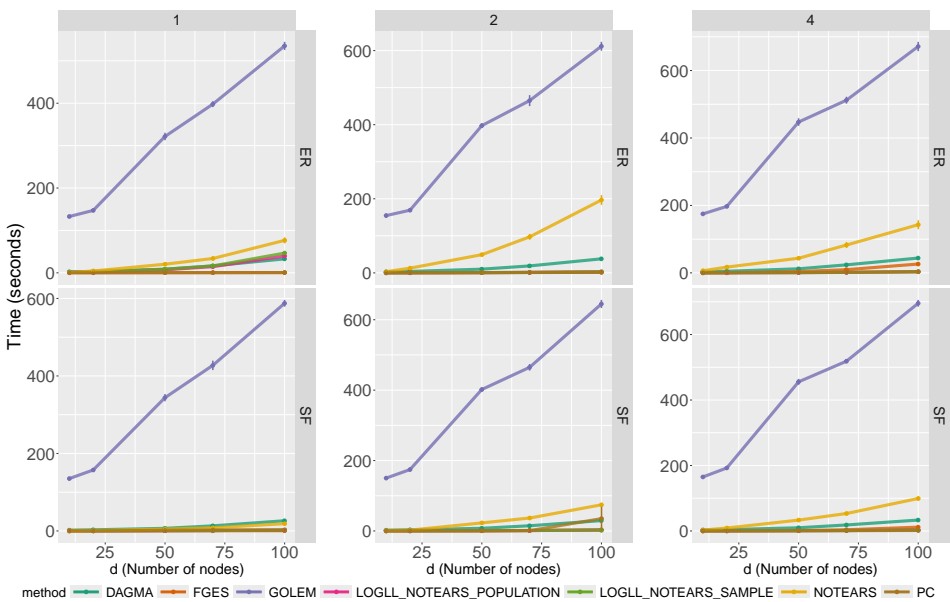

Figure 11: Results in term of Time. Lower is better. Column: $k = \{1, 2, 4\}$. Row: random graph types. {ER,SF}-$k$ = {Scale-Free,Erds-Rényi } graphs with $kd$ expected edges. Here $d = \{10, 20, 50, 70, 100\}$, $n = 1000$. Standard error is removed for better visualization. It is for different methods on standardized data $\mathbf{Z}$

### E.3 Nonlinear Model (SHD)

### E.3.1 Neural Network

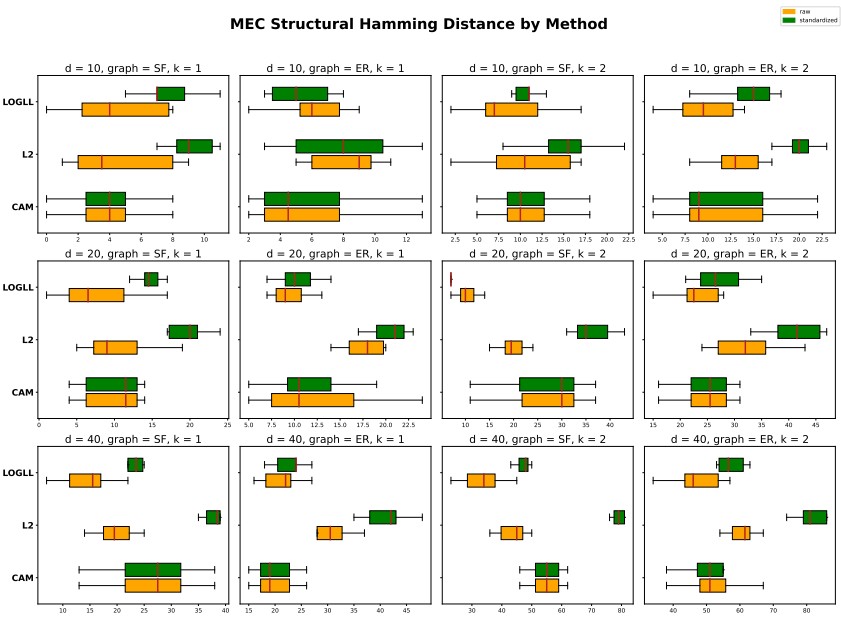

Figure 12: Structural Hamming distance (SHD) between Markov equivalence classes (MEC) of recovered and ground truth graphs. **LOGLL** (i.e. LOGLL-NOTEARS) stands for NOTEARS method with log-likelihood and quasi-MCP, **L2** (i.e. NOTEARS) stands for NOTEARS method with least square and $\ell_1$.

### E.3.2 General Linear Model with Binary Output (Logistic Model)

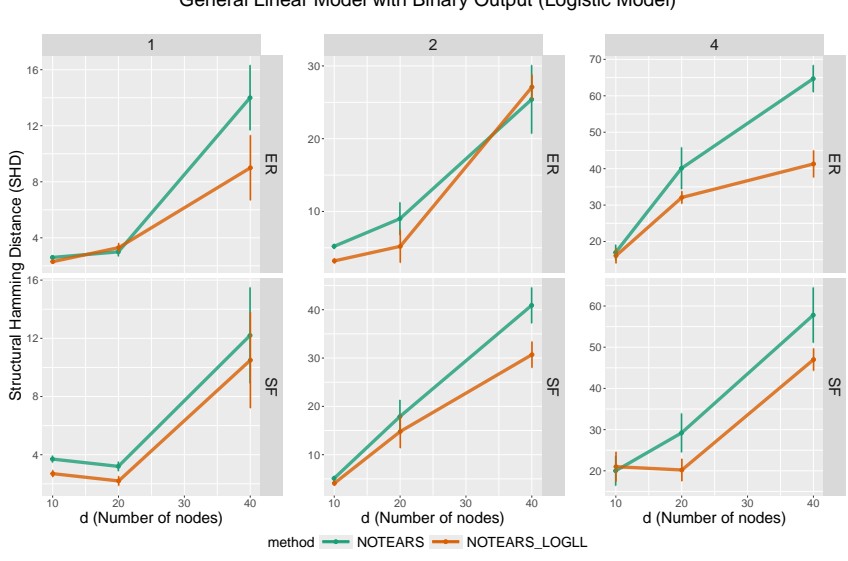

Figure 13: Structural Hamming distance (SHD) for Logistic Model, Row: random graph types, {SF, ER}-$k$= {Scale-Free,Erds-Rényi } graphs. Columns: $kd$ expected edges. NOTEARS_LOGLL (i.e. LOGLL-NOTEARS) uses log-likelihood with quasi-MCP, NOTEARS use log-likelihood with $\ell_1$. Error bars represent standard errors over 10 simulations.

