# OpenReview forum: "Markov Equivalence and Consistency in Differentiable Structure Learning"
_NeurIPS.cc/2024/Conference — NeurIPS 2024 poster_

### Official Review · Reviewer_mGjz · 2024-07-10

**Soundness:** 3
**Presentation:** 3
**Contribution:** 3
**Rating:** 6
**Confidence:** 4

**Summary:**

This paper proposed a differentiable DAG learning method using the log-likelihood loss and the minimax concave penalty (MCP). The author proved that under such construction, the minimizer of the loss identifies the sparsest graph (i.e. it has the minimal number of edges) which can generate the observational distribution. When faithfulness is further assumed, the identified graph is equivalent to the ground truth graph. The theorems are valid for linear Gaussian DAGs and general nonlinear SEMs where the induced distribution is parametric. The proposed method is validated by extensive experiments involving various dimensions and graph structures.

**Strengths:**

The paper is well-written and easy to follow.

The theoretical contribution is solid, demonstrating that a broad class of penalties (SCAD, MCD, quasi-MCD), along with the log-likelihood loss, identify the desired graph structure.

The simulations are well-designed and illustrate the outperformance of the proposed method.

**Weaknesses:**

The method can be regarded as a direct combination of several works. For example, the log-likelihood loss was also proposed in GOLEM [43]. The MCD penalty was first proposed in [72].

When applied to nonlinear models, the author used $\frac{1}{2n}\sum_{i=1}^d\log\left(||x_i-\hat f_i(X)||^2\right)$ as the log-likelihood loss. This seems only valid for homogeneous Gaussian noise where the variance in the denominator can be canceled out. However, in the experimental setting, the noise variance is heteroscedastic, making the likelihood improper.

Real-world datasets can be added to validate the proposed method.

Code is not available.

Some minor flaws in the presentation:
+ The conclusion section is missing.
+ Section D.3.2, time spent on the simulation. The plots are about SHD rather than about the running time.
+  [72] in the reference should be cited correctly.

**Questions:**

Can you provide further examples/illustrations on whether BIC or $L_1$ penalty can recover minimal models? I think this will strengthen the proposed method when faithfulness fails.

The log-likelihood for nonlinear models seems improper due to the heteroscedastic noise. Either the setting or the likelihood should be corrected before rebooting the experiments.

It would be interesting to compare the proposed method with the existing ones on real datasets.

**Limitations:**

The method assumes a known likelihood function, which can be impractical when the noise distribution or the functional structure is unknown.

---

> ### Author Rebuttal · Authors · 2024-08-06
>
> We would like to express our gratitude to the reviewer for their time, effort, and valuable suggestions. We would like to take this opportunity to address all the concerns raised in the reviews.
>
> > The method can be regarded as a direct combination of several works. For example, the log-likelihood loss was also proposed in GOLEM [43]. The MCD penalty was first proposed in [72].
> >
>
> Thanks for reviewer’s questions on our contribution. We would like to emphasize that our paper is not merely a combination of these previous works.
>
> (1) **New Identifiability Result:** We utilize our log-likelihood loss with a sparsity regularizer (MCP) to prove an identifiability result under common assumptions from the optimization literature. Unlike GOLEM [1], which has specific assumptions (such as the Triangle Assumption [1]) and relies on a linear model, our method does not require these conditions. Moreover, our proof is purely based on the properties of the score function and does not rely on any assumptions about the underlying graph structure, unlike GOLEM. Additionally, as far as we know, this is the first time MCP has been introduced in the continuous DAG learning literature. Previously, $\ell_1$ was the commonly used sparsity regularizer, which led to biased estimation of the underlying parameters and could negatively affect the recovered structure. By utilizing quasi-MCP, we overcome these drawbacks of $\ell_1$ and achieve unbiased estimation. Moreover, quasi-MCP also introduces significant technical challenges that our analysis handles (see also **Common Concern (3)**).
>
> (2) **Scale-invariant property:** We address a common concern for many DAG learning algorithms regarding scale invariance, wherein the algorithm may output different DAGs if the data is rescaled. To address this, we demonstrate that the log-likelihood score is scale-invariant under Gaussian noise. By showing this result, we aim to correct the common misunderstanding that the algorithm itself is affected by the scale of the data. Instead, it is the use of an incorrect score function that is sensitive to the scale.
>
> We believe these points distinguish our contributions from previous works, and are significant.
>
> > When applied to nonlinear models, the author used $\frac{1}{2n}\sum_{i=1}^d \log (\\|x_i -\hat{f}_i(x)\\|^2)$ as the log-likelihood loss. This seems only valid for homogeneous Gaussian noise where the variance in the denominator can be canceled out. However, in the experimental setting, the noise variance is heteroscedastic, making the likelihood improper.
>
> > The log-likelihood for nonlinear models seems improper due to the heteroscedastic noise. Either the setting or the likelihood should be corrected before rebooting the experiments.
>
> Thank you for this important question! We are happy to clarify the misunderstanding and will include the full derivations in the paper for clarity.
>
> When applied to a nonlinear model with heteroscedastic noise, the negative log-likelihood is:
>
> $$
> \frac{1}{2n}\sum\_{i=1}^d \log (\\|x_i -\hat{f}_i(x)\\|^2)
> $$
> However, when the noise is homogeneous, the negative log-likelihood becomes:
> $$
> \frac{d}{2}\log \left(\sum\_{i=1}^d \\|x_i - \hat{f}_i(x)\\|^2\right)
> $$
> Although these formulations appear similar, they are fundamentally different. With heteroscedastic noise, the loss is the log of the sum of least squares. In contrast, with homogeneous noise, the loss is the sum of the logs of least squares. The full derivation can be found in [1] (which covers both heteroscedastic and homogeneous noise in the linear case, with $\hat{f}_i(x) = B_i^\top x^{(i)}$) and [2] (which uses heteroscedastic noise in their setting).
>
> Thanks for this question! We address this point later in the Question “The log-likelihood….before rebooting the experiments.”
>
> > Real-world datasets can be added to validate the proposed method.
> >
>
> > It would be interesting to compare the proposed method with the existing ones on real datasets.
> >
>
> We include the real-data application in attached pdf (See **Common Concern (2)**). These results will be updated into our paper.
>
> > Code is not available.
> >
>
> Thank you for bringing this up, the code will be released in the camera ready.
>
> > The conclusion section is missing.
> >
>
> Thanks for pointing it out. Please refer to **Common Concern (4)** for conclusion and limitation.
>
> > Section D.3.2, time spent on the simulation. The plots are about SHD rather than about the running time.
> >
>
> Thanks for bring up this typo. We put the wrong name in the title and y-axis, it should be title: “$d$ vs Time for methods”, y-axis: "Time (second)”. The value in the plot is correct. This will be update in the paper correspondingly.
>
> > [72] in the reference should be cited correctly.
> >
>
> Thanks; this reference will be updated in the final version.
>
> > Can you provide further examples/illustrations on whether BIC or $\ell_1$ penalty can recover minimal models? I think this will strengthen the proposed method when faithfulness fails.
> >
>
> Thanks for the suggestion. Please refer to **Common Concern (2)** for more examples.
>
> [1] Ng, Ignavier, AmirEmad Ghassami, and Kun Zhang. "On the role of sparsity and dag constraints for learning linear dags." *Advances in Neural Information Processing Systems* 33 (2020): 17943-17954.
>
> [2] Bühlmann, Peter, Jonas Peters, and Jan Ernest. "CAM: Causal additive models, high-dimensional order search and penalized regression." (2014): 2526-2556.

---

> > ### Comment · Reviewer_mGjz · 2024-08-11
> >
> > Thank you for your response. Most of my concerns are addressed. At this moment, I would like to retain my score before further discussions with other reviewers.

---

> > > ### Author Response · Authors · 2024-08-12
> > > **Thank you!**
> > >
> > > Thank you for the reviewer’s comment. We hope you can have a more comprehensive assessment after discussing with the other reviewers. We appreciate your time and effort.

---

> > > > ### Comment · Reviewer_mGjz · 2024-08-13
> > > >
> > > > Dear authors,
> > > >
> > > > Thank you very much for your efforts in the discussion. Explicit examples and mathematical justification have shown that the model assumptions can incorporate general cases. Therefore, I will update my score to 6.

---

### Official Review · Reviewer_spWh · 2024-07-12

**Soundness:** 4
**Presentation:** 4
**Contribution:** 2
**Rating:** 5
**Confidence:** 4

**Summary:**

This paper introduces new identifiability results (MEC) based on maximum likelihood estimation complemented with sparsity regularization (quasi-MCP) for both a Gaussian linear model and a more general, potentially nonlinear models, under the very standard faithfulness assumption. The paper contains a theoretical analysis showing the score function is invariant to rescaling (in the Gaussian linear case only) and provide experiments showing the approach compares favorably to other baselines (in the Gaussian linear case only).

**Review summary:**
This paper reads very nicely and some of its proposals are new and interesting (scale-invariant result and the usage of the quasi-MCP to get MEC identifiability), but it missed a very relevant work [a] with very similar contributions (which significantly reduce the novelty factor of this manuscript) and a more complete set of experiments (see below). For these reasons, I believe this work is not ready yet for publication. That being said, I think it could get accepted at another conference by
- rewriting the paper in a more transparent way, especially by contrasting with the contributions of [b] (and changing the title);
- highlighting what are the actual contributions (scale-invariant result + MEC identifiability via quasi-MCP regularizer); and
- providing convincing experiments with nonlinear models.

**Strengths:**

- This paper is extremely well written and easy to follow. The notation is always well introduced with kind reminders when less standard notation is used.
- Overall the paper feels quite pedagogical.
- I believe the theoretical analysis of scale-invariance, although simple, is novel and very interesting, as it addresses problems raised with existing approaches regarding standardization.
- I think some aspect of the theoretical results is interesting, like the proof that the quasi-MCP regularizer can be used to get identifiability of the MEC. However, I believe very similar results have been shown previously in the literature (omitted from the related work) which seriously limit the novelty factor of this work.

**Weaknesses:**

**Important prior work omitted which seriously limits novelty**

The contribution is motivated by the need to consider more general score functions beyond the mean-squared error used in NOTEARS. The authors missed the work of [a] which proves identifiability for very general likelihood functions, including universal approximators (thanks to normalizing flows). The result of [a] applies to cases where interventions are observed, but covers also the special case without interventions, in which case the result guarantees identifiability of the MEC under the faithfulness assumption. Interestingly, [a] also requires the coefficient of the regularizer to be “small enough”, as is also the case in this manuscript. A key distinction with [a] is that this manuscript is analyzing a differentiable nonconvex sparsity regularizer as opposed to an L0 regularizer, which is not suitable for gradient-descent (in practice, [a] uses gumbel-sigmoid masks and regularizes their probability of being one). Am I missing other key distinctions? Showing that the quasi-MCP regularizer can yield MEC identification is new AFAIK and interesting IMO, but this combined with the simple scale-invariance results makes for a rather limited contribution...

I believe the title reflects the lack of knowledge of the existing literature on continuous structure learning. The title is too general. It sounds like the paper introduces likelihood-based differentiable structure learning, but that’s clearly not the case. [a] is a clear example with very similar theory (although not exactly encompassing this manuscript) as well as [b] and [c] (although without identifiability theory).

**Experiments investigate only the linear case**

The paper is largely motivated by the need to go beyond linear Gaussian score functions, but the experiments fall completely short of that promise, as only the Gaussian linear case is investigated. In contrast, [a] train neural network based architecture, with similar theoretical guarantees. More experiments are needed to confirm that the quasi-MPC regularizer transfers to nonlinear setting.

**Limitations of the theory:**

Assumption A: Assuming that the cardinality of equivalence of parameters is finite feels like a strong assumption. For instance, if you’d like to parameterize the conditional using neural networks, this assumption wouldn’t be satisfied since most architecture have infinitely many equivalent parameterization (think of ReLU activation where you can rescale the output of a neuron and undo this rescaling in the following layer). The result of [a] does not make this assumption despite their very similar setup. Why is it needed here? Also, can you give a concrete example of function B(\psi) that is Lipschitz.

Assumption B: The same example with ReLU neural networks is a counterexample to this assumption. Indeed, because of this rescaling, the set of parameters yielding the same network is unbounded. Excluding NN feels limiting, especially since existing results can cover this case.

**Unclear limit statements in Theorem 1 to 4**

The usage of the limit symbol “->” is a bit strange. Usually, we say that a_n -> b as n -> \infty, but here the result says a_n = b as n -> \infty. What does the latter even mean? Does that mean there exists a large enough N such that a_n = b for all n > N? I don’t think it’s true anyway. Why not presenting the result directly for the population likelihood (i.e. with \ell instead of \ell_n), as was done in [a]? Skimming the proof very quickly suggests that’s what was proved anyway. Note that this is different from showing consistency of the procedure, i.e. that your estimator converges to the right MEC as the number of samples grows (I suspect the latter will be more difficult to show)

Minor:
- It would be nice to have a small plot of the quasi-MCP regularizer.
- Typo in Lemma 3 in appendix? I think an \mathcal{M} should be replaced by a \mathcal{G}, no?
- Text is absurdly small in Figure 1 e.g.. Please fix this.
- Is the sparsest Markov representation always unique? The phrasing used suggests it is, but it might be worth expliciting.

[a] P. Brouillard, S. Lachapelle, A. Lacoste, S. Lacoste-Julien, and A. Drouin. Differentiable causal discovery from interventional data. In Advances in Neural Information Processing Systems, 2020.

[b] X. Zheng, C. Dan, B. Aragam, P. Ravikumar, and E. Xing. Learning sparse nonparametric dags. In Proceedings of the Twenty Third International Conference on Artificial Intelligence and Statistics, 2020.

[c] S. Lachapelle, P. Brouillard, T. Deleu, and S. Lacoste-Julien. Gradient-based neural DAG
learning. In Proceedings of the 8th International Conference on Learning Representations,
2020.

**Questions:**

See above.

**Limitations:**

This paper lacks a conclusion/discussion, so the limitations of the approach are not addressed.

---

> ### Author Rebuttal · Authors · 2024-08-06
>
> We sincerely thank the reviewer for the insightful critiques and comprehensive understanding of our work, and for providing such useful feedback. We will try our best to address the reviewer’s concern.
>
> > **Comparison to previous work**
> >
>
> We thank the reviewer for highlighting this related work; we will certainly update our paper to cite it and provide a discussion. Although our work and [1] prove similar results on identifiability through optimizing the log-likelihood with sparsity regularization, there are important differences. Most importantly, the use of $\ell_0$ significantly simplifies the analysis (similar identifiability results as Theorem 1, 2, and 4 are easily obtained, see **Common Concern (3)** for details) and ultimately does *not* lead to a differentiable program. Our focus is on a fully differentiable formulation. In fact, this is precisely what leads to Assumptions A and B, which are *not* needed when $\ell_0$-regularization is used. This is not surprising since $\ell_0$ leads to a combinatorial optimization problem which we are trying to avoid. Again, see **Common Concern (3)** for details.
>
> Moreover, our results rely on slightly weaker assumptions to get identifiability results, i.e., Sparsest Markov Representation (SMR) assumption, which has been shown to be weaker than faithfulness (with only observational data, $\mathcal{I}^*$-faithfulness assumption is regular faithfulness assumption [1]). Our proof is entirely different and more straightforward, relying solely on the score function, without the consideration of different skeletons and immoralities [1]. Our paper also includes new results on scale-invariance to address current concerns with many DAG learning algorithms.
>
> > **Experiments investigate only the linear case**
> >
>
> Originally, due to space limits, we put the nonlinear experiments results in Appendix D.3.3. (Page 31), which may have been easy to miss. Nonetheless, we have added more nonlinear experiments; please see **Common Concern (2)**.
>
> > **Limitations of the theory**
> >
>
> Thanks for these useful comments! Assumptions are needed because the our penalty term is quasi-MCP, introducing fundamental difficulties compared to previous work [1] using $\ell_0$. If penalty term is $\ell_0$, all these assumptions can be removed, and similar results hold. Please refer to **Common Concern (3)** for more details.
>
> > **Also, can you give a concrete example of function B(\psi) that is Lipschitz.**
> >
>
> First, in a (general) linear model, $\psi = B$. In this sense, $B(\psi) = \psi = B$, and $B(\psi)$ is 1-Lipschitz. Another example can be found in **Common Concern (1).** In this case, $\psi = \\{\beta^S\\}_{S\subseteq [p]}$, $[B(\psi)]\_{ji} = \sum\_{\{S:j\in S, i\in S, S\subseteq A\}}(\beta^{S})^2$, which is Lipschitz.
>
> Moreover, [2] uses a neural network to model $f_j$ in Equation (2). Let $\psi_j$ be the weights that connect the input layer and the first hidden layer in the $j$-th neural network. In this case, $\psi = (\psi_1, \ldots, \psi_p)$. It turns out that $[B(\psi)]\_{ij} = \\|\text{i-th-column}(\psi_j)\\|_2$. As a consequence, $\\|B(\psi)\\|\_2 = \sum\_{i,j} \\|\text{i-th-column}(\psi_j)\\|_2 \leq p \sum_j \\|\psi_j\\|_2 \leq p^2 \\|\psi\\|_2$. Therefore, it is $p^2$-Lipschitz. In general, $B(\psi)$ is a function of $\psi$, and usually, the nonzero part of $\psi$ indicates that a certain entry of $B(\psi)$ is nonzero. $\ell_1$ or $\ell_2$ norms are used to ensure nonzeros are mapped to nonzeros and zeros are mapped to zeros, which makes $B(\psi)$ inherit the Lipschitz property of $\ell_1$ or $\ell_2$.
>
> > **Unclear limit statements in Theorem 1 to 4**
> >
>
> Thank you for the suggestion, and we apologize for any confusion. Indeed, our results apply to the population case, which we indicate by using $n \rightarrow \infty$. For Theorems 1 to 4, we are stating that when considering the population case (infinite number of samples), certain equality relationships hold. We will revise these statement to make this more clear.
>
> > **Minor**
> >
>
> (1) Originally, we include the plot of quasi-MCP, but it is removed due to space limit. But it seems it will be better idea to get it back again.
>
> (2) Yes, this is a typo, it should be $\mathcal{M}(G^0) = \mathcal{G}(\mathcal{E}_{\min}(\psi^0,\xi^0))$
>
> (3) Good suggestion! it will be fixed.
>
> (4) Sparsest Markov representation(SMR) is not unique itself in general, but it is unique up to Markovian class.
>
> > Limitation
> >
>
> Thanks for bring it up. We provide conclusions and limitations in **Common Concern (4).**
>
> [1] Brouillard, Philippe, et al. "Differentiable causal discovery from interventional data." *Advances in Neural Information Processing Systems* 33 (2020): 21865-21877.
>
> [2] Zheng, Xun, et al. "Learning sparse nonparametric dags." *International Conference on Artificial Intelligence and Statistics*. Pmlr, 2020.

---

> > ### Comment · Reviewer_spWh · 2024-08-13
> >
> > I thank the authors for engaging with my review and providing a convincing rebuttal. The additional discussion contrasting their theoretical result with that of [1] was enlightening and very clear. I urge the authors to integrate these discussions in the next version of their work to better contextualize their contribution within existing literature. The authors should also consider changing the title to something less general, one way to do that would be to mention the quasi-MCP regularizer somehow (this paper is not the first differentiable likelihood based structure learning approach proposed!). Also, please remove the limit statement as these are inaccurate and should be replaced by "population results" (\ell instead of \ell_n).
> >
> > Assuming these changes will be properly addressed in the next revision, I raise my score from 3 to 5. I do not raise further as I still believe the novelty factor fairly low.

---

> > > ### Author Response · Authors · 2024-08-13
> > > **Thanks!**
> > >
> > > Dear Reviewer,
> > >
> > > We sincerely appreciate your positive feedback on our rebuttal. We will incorporate [1] and related works into the paper and expand the related work section to discuss the similarities and differences between our work and [1].
> > >
> > > We acknowledge that the current title is too general, and we will refine it to better reflect the paper’s content. Additionally, all statements involving $n\rightarrow \infty$ will be revised to refer to "population" results.
> > >
> > > Thank you once again for your invaluable suggestions and the time you’ve dedicated to reviewing our work.

---

### Official Review · Reviewer_o9h2 · 2024-07-12

**Soundness:** 4
**Presentation:** 3
**Contribution:** 3
**Rating:** 7
**Confidence:** 4

**Summary:**

The authors analyze a framework of sparsity-regularized maximum likelihood learning under the NOTEARS constraint for score-based causal discovery. Drawing from the sparsest permutations principle (a hybrid method), they show that a sparsity regularized likelihood objective is able to recover an element of the MEC even under structural non-identifiability of the SCM. The (non-identifiable) linear Gaussian setting is analyzed in detail, where it is shown that the MCP and SCAD penalties are able to recover the sparsest graph with appropriate hyperparameter settings. The authors then extend this to general likelihoods under the assumption that the (parameter) non-identifiability class (at the ground truth) is finite and other regularity conditions. For the linear-Gaussian case, the authors also prove scale invariance of the structure obtained from the method, addressing concerns of varsortability that similar methods are susceptible to. Finally, an experimental study is conducted on simulated linear and nonlinear ground truths.

**Strengths:**

I advocate for the acceptance of this paper based on two simple dimensions:

- (Significance and Motivation): Structural identifiability is a near-universal, yet entirely convenience-driven assumption when using likelihood-based scores for causal discovery. This work theoretically justifies NOTEARS-type approaches under structural non-identifiability, which broadens the class of applicable likelihood-based scores.

- (Clarity and Execution): I found the paper easy to follow, with precise mathematical notation, careful proofs and comprehensive analysis of the proposed framework.

**Weaknesses:**

However, I do think the paper falls somewhat short in a few (fixable) areas. Actionable questions are __bolded__, and I may raise my score if these points are clarified.

- The section on scale invariance seems out of place and somewhat weaker than the rest of the paper. It's clear that the Gaussian likelihood is scale invariant, and structure invariance is not a terribly surprising conclusion of the regularized objective also. __Unless the authors can clarify the contribution of this section, I feel like this is better off stated as a short paragraph or remark.__
- The authors state that the finite parameter equivalence class assumption in the general case is "relatively mild", and that it is satisfied by a range of models ("most exponential families"). I'm not convinced that this is the case. First, typically when statistical models with continuous parameter spaces are non-identifiable, the equivalence class is usually infinite. For example, I believe exponential families in canonical parametrization are either full-rank, where the parameter is identifiable (and hence not relevant to the motivation of the paper), or non-identifiable up to entire subspaces of the parameter space (Theorem 1 of [1]), which are infinite. __Could the authors clarify what they mean on l297-298?__
- The authors do not provide any examples, or references, of non-Gaussian log-likelihoods scores that would be useful for causal discovery. The non-linear example in the experiments seems to use a MSE loss which is the same as non-linear NOTEARS [2], and the only improvement shown in the experiments is due to changing the optimization scheme to Adam (and adding the quasi-MCD regularizer?). __Could you provide (practical) examples of likelihoods where the theory in Section 5 hold? I.e., other non-identifiable likelihood scores that are useful for causal discovery?__.

[1] Notes by Charles Geyer: https://www.stat.umn.edu/geyer/8053/notes/expfam.pdf

[2] "Learning Sparse Nonparametric DAGs", Zheng et al., AISTATS 2020.

**Questions:**

### Other questions

- You mention that $\ell_1$ loss is "not effective" compared to MCP, SCAD, is this a computational or theoretical issue? In other words, do you have a conjecture/result on whether the theoretical claims no longer hold when replaced with $\ell_1$?
- Although the likelihood is "structurally" scale-invariant (Thm. 3), the scale presumably can still affect (uniform over d) thresholding in finite samples by changing the speed at which structural zeros converge. This has always been my intuition on how var-sortability works, and not that the MSE in NOTEARS is actually "structurally" affected by scaling. Is there a reference to show that the original MSE loss is not also structurally scale-invariant? If the MSE is also structurally scale-invariant, do you conjecture that the likelihood has desirable properties in terms of thresholding?

**Limitations:**

The authors do not adequately discuss the limitations of the paper in my opinion (in fact, the paper abruptly ends without even a conclusion), and this is also seen in in the checklist where the justification for this section was left blank.

I think the paper deserves a proper conclusion and discussion of the (potential) limitations of Assumption A1 in Section 5. __I do not think this is a tough ask, so I may lower my score if this remains unaddressed in the rebuttal.__

---

> ### Author Rebuttal · Authors · 2024-08-07
>
> We would like to express our gratitude to the reviewer for acknowledging the value of our contributions and clarity of presentation. All the points addressed below will be updated in our paper.
> > The section on scale invariance ….  or remark.
>
> We will move these details to the appendix, and shorten the discussion in the main paper.
>
> > The authors state that the finite parameter…..mean on l297-298?
>
> First, consider simple case of Gaussian models, and then below we discuss how to generalize. Let $X \sim \mathcal{N}(0, \Sigma)$ and $\Sigma\succ0$. It belongs to the exponential family and is full-rank. The covariance $\Sigma$ is identifiable, however, we are interested in the matrix $B$ in the SEM $X = B^\top X + N$. As in Section 4.1, we know that $\Sigma = (I - B)^{-\top} \Omega (I - B)^{-1}$. As we discuss in Section 4.2, there are at most $p!$ different $B$ that satisfy this condition, making our problem **unidentifiable with respect to $B$, but with a finite number of equivalent parameters**.
>
> **As for unidentifiable exponential families in [4],** we can always reduce it to an identifiable model by projecting canonical statistics to a subspace. For the same example, if $\Sigma$ is singular, we can decompose $X = (X_a, X_b)$ such that $\text{Cov}(X_a)\succ 0$  and has the same rank as $\Sigma$, and $X_b$ is a linear combination of $X_a$. As discussed before, there is a finite equivalence class for $X_a$, and $X_b$ has a deterministic relationship to $X_a$.
>
> Another explicit example (GLM with binary data) can be found in the **Common Concern (1)**. This also extends to more general exponential families. When exponential families are full-rank, although their parameters are identifiable, the parameters $(\psi, \xi)$ as well as the SEM $B(\psi)$ may not be identifiable, which are our interest. As before, parameter is $\Sigma$, $\psi = B, \xi = \Omega$, and $\Sigma$ is function of $(B,\Omega)$. Moreover, in many cases, the size of $\mathcal{E}(\psi^0, \xi^0)$ is closely related to $p!$, which is finite since there are $p!$ topological sorts $\pi$. For each $\pi$, it is usually associated with a pair $(\psi(\pi), \xi(\pi))$ in a certain way such that $P(x; \psi(\pi), \xi(\pi)) = P(x; \psi^0, \xi^0)$. This is why we claim Assumption A(1) is reasonable.
>
> Regarding Assumption A and B, more discussion can be found in the **Common Concern (3).**
>
> > The authors do not provide…scores that are useful for causal discovery?.
>
> The point is that *any* likelihood can be used in conjunction with the quasi-MCP. Two examples of non-Gaussian likelihoods are logistic (binary) models and additive noise models. A third example is any model for the joint distribution with GLMs for CPDs. All told, these comprise a very rich class of models with explicitly derivable score functions. Unfortunately, we have not made this point explicit and will certainly do so in the camera ready.
>
> For additive noise models, the log-likelihood function is $\frac{1}{2n}\sum_{i=1}^d \log (\\|x_i -\hat{f}\_i(x)\\|^2)$[1], which is used in our experiments (see Line 855 in Appendix). It is different from MSE loss $\frac{1}{2n}\sum_{i=1}^d\\|x_i - \hat{f}_i(x)\\|_2^2$ used in NOTEARS [2]. For the improvement of the experiments, it is based on the correct use of loss function and penalty, thus it leads to better performance. For other examples, please refer to **Common Concern (1).**
>
> > You mention that $\ell_1$ loss ... when replaced with $\ell_1$?
>
> Using $\ell_1$ as a penalty can lead to biased parameter estimates because it generally shrinks coefficients to zero. In contrast, MCP provides unbiased estimates. Specifically, with MCP, $\mathcal{O}_{n,\lambda,\delta} = \mathcal{E}\_{\min}(\Theta)$ when population case is considered. However, this is not true for $\ell_1$.
>
> Consider $X_1\sim \mathcal{N}(0,1)$, $X_2\sim X_1+\mathcal{N}(0,1)$, with the known order ($X_1\rightarrow X_2$), so that the log-likelihood with $\ell_1$ penalty is
> $$
> \log((1-a)^2+1)+\lambda|a|
> $$
>
> It is easy to see $a = 1$ is not minimizer. This indicates that $\ell_1$ leads to biased estimates.
>
> > Although the likelihood is "structurally"….. in terms of thresholding?
>
> Good point! Although scaling does not affect which entries in the adjacency matrix are zero, it changes the magnitude of the entries. With a finite number of samples, this alters the thresholding we should use, which significantly affect the recovered structure in practice.
>
> With respect to MSE loss, it depends: It is scale-invariant for a fixed topological sort, but this is not true in general when the sort is unknown. Let us clarify:
>
> 1. For fixed topo sort, the MSE in NOTEARS is scale-invariant. Since the matrix $B$ can be recovered by linear regression, and changing the scale of the data $X$ does not change the positions of the nonzero entries in $B$, leading to invariance. This is similar to the intuition of Theorem 3.
> 2. However, the MSE loss in NOTEARS is not scale-invariant when the sort is unknown. For instance, consider $X = (X_1,X_2)$ where $X_1 = N_1, X_2 =aX_1+ N_2, N_1, N_2\overset{i.i.d}{\sim}\mathcal{N}(0,1)$. Then the optimal for MSE loss is
> $$
> B^* = \begin{bmatrix} 0 & a \\\\ 0 & 0 \end{bmatrix}
> $$
> However, if you standardize $X$ to get $Z =(X_1,X_2/\sqrt{1+a^2})$ and optimal are
> $$
> B^*_1 = \begin{bmatrix}0&a/\sqrt{a^2+1} \\\\ 0&0\end{bmatrix}\qquad B^*_2 = \begin{bmatrix}0&0 \\\\ a/\sqrt{a^2+1}&0\end{bmatrix}
> $$
> So MSE is not scale-invariant in the sense that global optimal solution can not have the same structure for different scale of data. However, this is not the case for likelihood (Theorem 3).
>
> > Limitation and conclusion
>
> Please refer to **Common Concern (4)**. More discussions on Assumption A1 can be found in previous response. We will add this to the paper with the extra page in the camera ready.
>
> [1] Bühlmann, et al "CAM: Causal additive models, high-dimensional order search and penalized regression" (2014)
>
> [2] Zheng, et al. "Learning sparse nonparametric dags" (2020)

---

> > ### Comment · Reviewer_o9h2 · 2024-08-12
> >
> > Thank you for the response. I will retain my score for now, to discuss with other reviewers, who also found the equivalence classes confusing. Based on your $p!$ characterization, it seems equivalence classes contain either DAGs, or parameter settings in 1-to-1 correspondence with DAGs. Upon reading C.2., is it the case that for a given sparsity pattern (i.e., DAG), only one B setting in the Gaussian example will yield the same distribution? I was originally thinking that rotations gave infinitely many equivalent parameters (and this generalizes in EFs) but this would break the sparsity. Could the authors comment on whether this understanding is correct?
> >
> > Note this was not at all clear to me on the first few passes so I would appreciate if you could put a version of C.2 in the main text.
> >
> > > As we discuss in Section 4.2, there are at most $p!$ different $B$ that satisfy this condition
> >
> > I'm wondering if you mean C.2? I can't actually find this statement anywhere.

---

> ### Author Response · Authors · 2024-08-12
> **Thank you for valuable comment!**
>
> Thank you for the reviewer’s further comments, and we apologize for the unclear part. We are happy to address any concerns you have and ensure a clear understanding of our work. Please rest assured, we will incorporate all the suggestions into the paper, including moving Section C.2 to the main paper to provide readers with more background knowledge.
>
>
>
> > Is it the case that for a given sparsity pattern (i.e., DAG), only one $B$ setting in the Gaussian example will yield the same distribution?
> >
>
> You are correct! From my understanding, if a sparsity pattern here refers to a topological sort $\pi$. For each $\pi$, there will be **only one** $B$ that is consistent with $\pi$ and can generate the distribution.
>
> Consider a simple three-node example. Let $X \sim \mathcal{N}(0, \Sigma)$, where $\Sigma \succ 0$:
> $$
> \Sigma = \begin{bmatrix}
> 1/2&1&1\\\\1&3&3\\\\1&3&4
> \end{bmatrix}
> $$
> We would like to recover $B$ (DAG) such that $X = B^\top X + N$ and $X \sim \mathcal{N}(0, \Sigma)$. In this case, there are $p! = 3! = 6$ different topological sorts, and at most six different $B$s can generate $X$. Based on Section C2, **it is worth noting that aside from these different $B$s, no other $B$ can generate the distribution $X$ [1]**. These $B$s and $\pi$s are:
> $$
> B(\pi_1) = \begin{bmatrix}
> 0&0&0\\\\1/3&0&0\\\\0&3/4&0
> \end{bmatrix}\qquad \pi_1 = [2,1,0]
> $$
> $$
> B(\pi_2) = \begin{bmatrix}
> 0&0&0\\\\1/3&0&1\\\\0&0&0
> \end{bmatrix}\qquad \pi_2 = [1,2,0]
> $$
> $$
> B(\pi_3) = \begin{bmatrix}
> 0&1&0\\\\0&0&0\\\\1/4&1/2&0
> \end{bmatrix}\qquad \pi_3 = [2,0,1]
> $$
> $$
> B(\pi_4) = \begin{bmatrix}
> 0&1&2\\\\0&0&0\\\\0&1/2&0
> \end{bmatrix}\qquad \pi_4 = [0,2,1]
> $$
> $$
> B(\pi_5) = \begin{bmatrix}
> 0&0&0\\\\1/3&0&1\\\\0&0&0
> \end{bmatrix}\qquad \pi_5 = [1,0,2]
> $$
> $$
> B(\pi_6) = \begin{bmatrix}
> 0&2&0\\\\0&0&1\\\\0&0&0
> \end{bmatrix}\qquad \pi_6 = [0,1,2]
> $$
> Here $B(\pi_5) = B(\pi_2)$, although $\pi_2\ne \pi_5$. This is why we say it at most $p!$ different $B$.
>
> For unidentifiable exponential families, consider the case where $X \sim \mathcal{N}(0,\Sigma)$ and $\Sigma$ is singular. For example,
>
> $$
> \Sigma = \begin{bmatrix}
> 1&1&1\\\\1&1&1\\\\1&1&2
> \end{bmatrix}
> $$
>
> We can reduce this to a lower dimension. Note that here $X_1 = X_2$, so we only need to consider $(X_2, X_3)$:
>
> $$
> \text{Cov}(X_2,X_3) = \begin{bmatrix}
> 1&1\\\\1&2
> \end{bmatrix}\succ0
> $$
>
> Then it reduces to previous case.
>
> As for more general model, for each $\pi$, it is usually associated with **only one** pair $(\psi(\pi), \xi(\pi))$ in a certain way such that $P(x; \psi(\pi), \xi(\pi)) = P(x; \psi^0, \xi^0)$, similar to Linear Guassian case. Another example in **Common concern (1)** follows such property.
>
> > I'm wondering if you mean C.2? I can't actually find this statement anywhere.
> >
>
> Apologies for the misunderstanding. You are correct; it should be Section C.2. We will include more details in Section 4.2 for clarity!
>
> As stated in Section C.2, for any topological sort $\pi$, it corresponds to a pair $(\tilde{B}(\pi), \tilde{\Omega}(\pi)) \in \mathcal{E}(\Theta)$. For all DAGs with $p$ nodes, there are at most $p!$ different topological sorts. This explains why there are at most $p!$ different $B$ matrices that satisfy this condition.
>
> [1] Aragam, Bryon, and Qing Zhou. "Concave penalized estimation of sparse Gaussian Bayesian networks." *The Journal of Machine Learning Research* 16.1 (2015): 2273-2328.

---

> > ### Comment · Reviewer_o9h2 · 2024-08-12
> > **Thank you!**
> >
> > Dear Authors, thank you for the clarification. This addresses my main concern, so I will raise the score to 7 assuming it is clarified in the paper.

---

> > > ### Author Response · Authors · 2024-08-12
> > > **Thank you so much!!!**
> > >
> > > We sincerely appreciate the time and effort you put into reviewing our paper and providing valuable feedback. We will incorporate your suggestions to enhance the clarity and strength of our work for our readers!

---

### Author Rebuttal · Authors · 2024-08-06

**(1) Another example beyond the linear gaussian model.**
Note that Assumption A holds as long as $P(X_i|X_A)$ has a unique SEM parametrization for any $i$ and $A$, where $A\subseteq[p]\backslash i$.  Because for any fixed topological sort $\pi$, we could let $A = \\{\text{parents node of }i \text{ in }\pi \\}$, and this results in at most $p!$ equivalence class. This is for example true in the Gaussian model.

Another example is a generalized linear model with binary output, i.e., $X = (X_1,\ldots,X_p)$ and $X_i\in\\{0,1\\}$ for $i = 1,\ldots,p$. Let $B = (B_1,\ldots,B_p)$, then  $\mathbb{E}[X_i\mid X_{pa(i)}] = g(B_i^\top X)$ where $g(s) = e^s/(1+e^s)$ which is equivalent to the following SEM

$$
X_i = \text{Bernoulli}(\exp(B_i^\top X)/(1+\exp(B_i^\top X)))\qquad i = 1,\ldots,p
$$

In this case, it can be shown that

$$
X_i\mid X_A \sim \text{Bernoulli}\Big(\frac{1}{1+\exp(- \sum_{r =1}^{|A|+1}(\sum_{j_1 = i,j_2,\ldots,j_r\in A}\beta^{ij_2\ldots j_r}x_{j_2}\ldots x_{j_r}))}\Big)
$$

is the unique parametrization of the conditional distribution (in terms of the coefficients $\beta^{ij_2\ldots j_r}$, which are a function of $B$). Thus, binary models also satisfy Assumption A. We will add this example to the paper.

**(2) More experiment on Nonlinear Neural Network, and General Linear model with binary output**

We conducted more experiments on Nonlinear Neural Networks (see Appendix D for details), and generalized linear models with binary output. The results are attached in the pdf.

**(3) Assumption A, B for Theorem 4**

First of all, we would like to emphasize that when using $\ell_0$ as the penalty term, Assumptions A and B are not needed at all. The proof for this case would be significantly simplified (see below for a proof sketch). Replacing $\ell_0$ with the differentiable quasi-MCP introduces significant complications, requiring some additional assumptions that distinguish our work from [1]. In fact, our assumptions are exactly what is needed to make the problem amenable to gradient-based optimization.

Considering Assumption A, the finiteness can be relaxed. What is really needed is that the minimal nonzero edge has enough “signal,” i.e.,

$$
\min\_{(\psi,\xi)\in \mathcal{E}(\psi^0,\xi^0)}\min\_{\\{(i,j):B(\psi)_{ij}\ne 0\\}}|B(\psi)|\_{ij}>0
$$

This is trivially true when $|\mathcal{E}(\psi^0,\xi^0)|$ is finite. When it is infinite, each $|B(\psi)|\_{ij}$ could be positive, but it is possible $\lim\inf_{(\psi,\xi)\in \mathcal{E}(\psi^0,\xi^0)}\min\_{\\{(i,j):B(\psi)_{ij}\ne 0\\}}|B(\psi)|\_{ij}=0$, because $|B(\psi)|\_{ij}$ can be arbitrarily small. The $\ell_0$ penalty deals with this with its discontinuity at zero, whereas the continuity of quasi-MPC makes this more challenging. This is the cost of differentiability, which we argue is worthwhile.

As for Assumption B, this is a common assumption in the optimization literature and quite weak in general. Moreover, this is nearly necessary because quasi-MCP cannot count the number of edges in $B(\psi)$ exactly: The magnitude of the quasi-MCP penalty does not reveal the number of edges. This is the price to pay when we replace $\ell_0$ with a fully differentiable sparsity penalty. Finally, we can point out that this can also be relaxed: What is needed is that for any $\epsilon>0,$ there exists  $\delta>0$

$$
\ell(\psi,\xi)-\ell(\psi^0,\xi^0)>\delta\quad \text{for }\\{(\psi,\xi)\mid \text{dist}((\psi,\xi),\mathcal{E}(\psi^0,\xi^0))>\epsilon\\}
$$

In other words, it requires a loss gap when $(\psi,\xi)$ is not in $\mathcal{E}(\psi^0,\xi^0)$.

Finally, we include a brief proof sketch of a similar result to Theorem 4 when $\ell_0$ is used. This illustrates that the quasi-MCP indeed introduces fundamental difficulties. Moreover, the reason the proof simplifies compared to [1] is a) The proof below does not consider interventions (which is a major innovation of [1]), and b) The use of the weaker SMR assumption (vs faithfulness) simplifies the analysis.

**Proof sketch:** We can also assume $|\mathcal{E}\_{\min}(\psi^0,\xi^0)|=1$, following the same reason as for Theorem 4.

When $s_{B(\psi^0)} = 0,$ the result is obvious. When $s_{B(\psi^0)}>0$ , divide the parameter space into $A_1 = \\{(\psi,\xi)\mid s_{B(\psi)} > s_{B(\psi^0)}\\}, A_2 = \\{(\psi,\xi)\mid s_{B(\psi)} = s_{B(\psi^0)}\\}, A_3 = \\{(\psi,\xi)\mid s_{B(\psi)} < s_{B(\psi^0)}\\}$. It is straightforward to verify each case. For example, for $A_2$, since $|\mathcal{E}\_{\min}(\psi^0,\xi^0)|=1$, that implies $\forall(\psi,\xi)\in\mathcal{E}(\psi^0,\xi^0)$ and  $(\psi,\xi)\ne (\psi^0,\xi^0)$, it holds $s_{B(\psi)}>s_{B(\psi^0)}$. Therefore, $\forall (\psi,\xi)\in A_2$, it holds $\ell(\psi^0,\xi^0)<\ell(\psi,\xi)$. As consequence, for any $\lambda>0$, result holds.

The other cases are similar with slight modifications.

(4) **Missing conclusion and limitation:** We proposed a fully differentiable score function for causal discovery, composed of log-likelihood and quasi-MCP. We demonstrated that the global solution corresponds to the sparsest DAG structure that is Markov to the data distribution. Under mild assumptions, we conclude that all optimal solutions are the sparsest within the same Markov Equivalence Class. Additionally, the proposed score is scale-invariant, producing the same structure regardless of the data scale under the linear Gaussian model. Experimental results validate our theory, showing that our score provides better and more robust structure recovery compared to other scores.

However, there are limitations to our work. We focus on parametric models and rely on assumptions such as the finiteness of the equivalence class and the boundedness of the level set of the log-likelihood. These assumptions limit the applicability of our theorem. Future work should explore ways to relax these assumptions to extend our method’s applicability to broader scenarios.

[1] Brouillard, et al. "Differentiable causal discovery from interventional data." (2020)

---

### Decision · Program_Chairs · 2024-09-25

**Decision:**

Accept (poster)

**Comment:**

This paper presents a fully-differentiable approach for DAG structure learning based on the log-likelihood loss and quasi-MCP (minimax concave penalty) regularization. The main contributions are in terms of identifiability results (the method identifies the sparest graph that is Markov to the data distribution) and scale invariance properties (the proposed method produces the same DAG regardless of the data scale).

Several concerns were raised during the discussion period regarding (1) experiments only investigating the linear Gaussian case; (2) the omission (and lack of discussion) of closely related work by Brouillard et al (NeurIPS 2020); (3) the limitations of the theory (3) and the lack of conclusions in the manuscript and discussion of limitations of the proposed approach. I believe the authors have provided a comprehensive rebuttal addressing the above and the reviewers have acknowledged that in their comments and scores updates.

Therefore, I recommend acceptance.